# Bypassing pan-enterovirus host factor PLA2G16

Jim Baggen[1,4,5], Yue Liu[2,4,5], Heyrhyoung Lyoo [1], Arno L.W. van Vliet[1], Maryam Wahedi[1], Jost W. de Bruin[1], Richard W. Roberts[1], Pieter Overduin[3], Adam Meijer[3], Michael G. Rossmann[2], Hendrik Jan Thibaut [1,4,5] & Frank J.M. van Kuppeveld[1,4,5]

Enteroviruses are a major cause of human disease. Adipose-specific phospholipase A2 (PLA2G16) was recently identified as a pan-enterovirus host factor and potential drug target. In this study, we identify a possible mechanism of PLA2G16 evasion by employing a dual glycan receptor-binding enterovirus D68 (EV-D68) strain. We previously showed that this strain does not strictly require the canonical EV-D68 receptor sialic acid. Here, we employ a haploid screen to identify sulfated glycosaminoglycans (sGAGs) as its second glycan receptor. Remarkably, engagement of sGAGs enables this virus to bypass PLA2G16. Using cryo-EM analysis, we reveal that, in contrast to sialic acid, sGAGs stimulate genome release from virions via structural changes that enlarge the putative openings for genome egress. Together, we describe an enterovirus that can bypass PLA2G16 and identify additional virion destabilization as a potential mechanism to circumvent PLA2G16.

[1] Virology Division, Department of Infectious Diseases and Immunology, Faculty of Veterinary Medicine, Utrecht University, 3584 CL Utrecht, The Netherlands. [2] Department of Biological Sciences, Purdue University, West Lafayette, IN 47907, USA. [3] Virology Division, Centre for Infectious Diseases Research, Diagnostics and Screening, National Institute for Public Health and the Environment, 3720 BA Bilthoven, The Netherlands. [4]These authors contributed equally: Jim Baggen, Yue Liu, Hendrik Jan Thibaut, Frank J. M. van Kuppeveld. Deceased: Michael G. Rossmann. Correspondence and requests for materials should be addressed to F.J.M.v.K. (email: f.j.m.vankuppeveld@uu.nl)

The *Enterovirus* genus (family *Picornaviridae*) comprises a large group of non-enveloped positive-stranded RNA viruses harboring 13 species, of which seven contain human pathogens (*Enterovirus A–D* and *Rhinovirus A–C*)[1]. Due to the absence of an envelope, enteroviruses need to deliver their genome across a membrane to infect a cell. Following receptor binding and endocytosis, virus particles uncoat by forming a proteinaceous pore spanning the endosomal membrane through which they eject their genome into the cytosol[2,3]. The uncoating process is triggered by cellular uncoating cues (receptor binding or low endosomal pH), which promote virion conversion into an expanded state (A-particle) that favors membrane interaction. Most enteroviruses, such as poliovirus and coxsackievirus B3 (CV-B3), require only their receptor for uncoating, whereas other viruses, like enterovirus A71 (EV-A71), also require acidification of the endosomal lumen[4]. Minor-group rhinoviruses (including rhinovirus A2 (RV-A2)) are exceptional, as these do not engage an uncoating receptor but uncoat exclusively via endosomal acidification[5]. Uncoating receptors bind to a depression on the outer surface of the virus called the canyon[6] and interact with the VP1 GH loop, thereby inducing the displacement of a stabilizing lipid, the pocket factor, from a hydrophobic pocket below the canyon[7].

Recently, using haploid genetic screens, PLA2G16 was identified as an enterovirus host factor[8,9]. PLA2G16 is a lipid-modifying enzyme[10,11] implicated in adipocyte lipolysis[12]. Knockout of PLA2G16 did not affect virus binding to the cell surface, endocytic uptake, translation or replication but hampered the dissociation of genomic RNA from virus-containing endosomes. This showed that PLA2G16 promotes the timely delivery of the viral genome to the cytosol, before the virus-induced membrane permeation is detected by the cellular sensor galectin 8 and the virus is cleared by the autophagy machinery[8]. The mechanism by which PLA2G16 facilitates viral genome delivery remains to be established.

EV-D68 is an atypical enterovirus that usually causes mild respiratory tract disease, but is also associated with severe lower respiratory tract infections and acute flaccid myelitis[13]. Because EV-D68 used to be a rare pathogen, it was poorly studied for a long time. However, the increased incidence of EV-D68-associated illness[14] has accelerated research on this virus. Using a genome-wide haploid genetic screen with the EV-D68 prototype strain Fermon, we previously identified sialic acid (Sia) as an essential host factor and showed that both α2,3- and α2,6-linked Sia can serve as receptors[15]. Moreover, crystallography revealed a Sia-binding site in the canyon and showed that Sia binding dislodges the pocket factor[16], suggesting a role of Sia in facilitating uncoating. However, no A-particle formation or genome release was observed upon incubation with Sia. Another study identified the neuron-specific protein intercellular adhesion molecule 5 (ICAM-5) as an EV-D68 receptor and showed that glycosylation at residue Asn[54] in ICAM-5 is required for virus binding[17]. ICAM-5 was found to promote the transformation of mature virions to A-particles and to facilitate genome release in vitro[17,18], implicating ICAM-5 as an EV-D68 uncoating receptor. Additionally, it was recently shown that EV-D68 infection of cells also requires endosomal acidification[19] and that acid treatment induces A-particle formation in vitro[20]. It is currently unknown how ICAM-5, Sia, and endosomal acidification cooperate to destabilize EV-D68 virions.

Although Sia is an essential receptor for most strains, we previously identified several EV-D68 clinical isolates, including EV-D68-947, that were able to infect Sia-deficient HAP1 SLC35A1[KO] cells[15], indicating that these strains can use alternative receptors. In this study, we set out to identify such alternative receptors and show that, in the absence of sialylated glycans, EV-D68-947 can employ sulfated glycosaminoglycans (sGAGs). We reveal that this dual receptor-binding strain requires PLA2G16 only when infecting cells via Sia, but not when using sGAGs. Using cryo-EM analysis, we show that sGAG binding induces structural rearrangements in the viral capsid and stimulates genome release in vitro. Our findings suggest a role of PLA2G16 as the last in a series of uncoating cues and point towards additional virion destabilization as a mechanism to circumvent the pan-enterovirus host factor PLA2G16.

## Results

**Both Sia and sGAGs can serve as EV-D68-947 receptors.** To ensure exclusive usage of non-sialylated receptors, we performed a haploid genetic screen with EV-D68-947 in HAP1 SLC35A1[KO] cells, which lack surface-expressed Sia. This screen identified many genes related to the synthesis of sGAGs, which are long, unbranched, highly sulfated polysaccharides (Fig. 1a, b). Hits include genes involved in synthesis of the GAG core tetrasaccharide (*B3GAT3, FAM20B, B3GALT6, B4GALT7, UXS1*, and *XYLT2*), elongation of heparan sulfate (*EXT1, EXT2,* and *EXTL3*), synthesis of UDP-glucuronate (*UGP2, UGDH*) and sulfation (*SLC35B2, NDST1*). Together, these hits pointed towards a role of sGAGs as an alternative receptor for EV-D68-947. To confirm that EV-D68-947 binds to sGAGs, we performed neutralization experiments using various soluble sGAG analogues. Virus incubation with soluble heparin or low-molecular-weight-heparin (LMWH) neutralized infection with EV-D68-947, while the Sia-dependent EV-D68-Fermon was not neutralized (Fig. 1c). Similarly, EV-D68-947 infection was neutralized by soluble heparan sulfate as well as fondaparinux (sulfated pentasaccharide) and dp6 (a heparin-derived hexasaccharide) (Supplementary Fig. 1a). To determine to what extent EV-D68-947 relies on Sia or sGAGs as a receptor, cells were treated with neuraminidase (NA), to remove Sia, or with sodium chlorate (NaClO$_3$), which prevents cell-surface sulfation. As previously described[15], EV-D68-947 was more resistant to NA treatment of HeLa cells (Fig. 1d) and HAP1 cells (Supplementary Fig. 1b) than EV-D68-Fermon. Treatment with NaClO$_3$ only partially inhibited infection, while combined treatment with NA and NaClO$_3$ inhibited EV-D68-947 to a similar extent as EV-D68-Fermon following NA treatment (Fig. 1d). Analogously, the combination of NA treatment and knockout of *B3GALT6*[21], which is required for sGAG synthesis, synergistically inhibited EV-D68-947 infection of HAP1 cells (Supplementary Fig. 1b). Treatment with chlorpromazine, an inhibitor of clathrin-mediated endocytosis, showed that engagement of Sia leads to clathrin-mediated virus internalization, whereas sGAG-mediated infection is not strictly dependent on this route (Supplementary Fig. 1c). Altogether, these data indicate that EV-D68-947 is a dual receptor-binding virus, using either Sia or sGAGs, and that the mode of virus internalization is receptor-dependent. EV-D68 has also been described to use the neuron-specific protein intercellular adhesion molecule 5 (ICAM-5) as an uncoating receptor[17]. However, *ICAM-5* was not among the haploid screen hits. We observed that HAP1 ICAM-5[KO] cells can be infected with EV-D68-947, albeit less efficiently (Supplementary Fig. 1d). These data suggest a possible redundancy between ICAM-5 and other (protein) receptors.

**Engagement of sGAGs reduces the PLA2G16 dependency of EV-D68.** PLA2G16 has been identified as a host factor in several enterovirus haploid genetic screens[8,15], including a screen with EV-D68-Fermon (Fig. 2a). In contrast to these screens, no significant enrichment of disruptive gene-trap insertions in *PLA2G16* was observed in the screen with EV-D68-947 in SLC35A1[KO] cells (Fig. 2a), suggesting that this strain has unusual

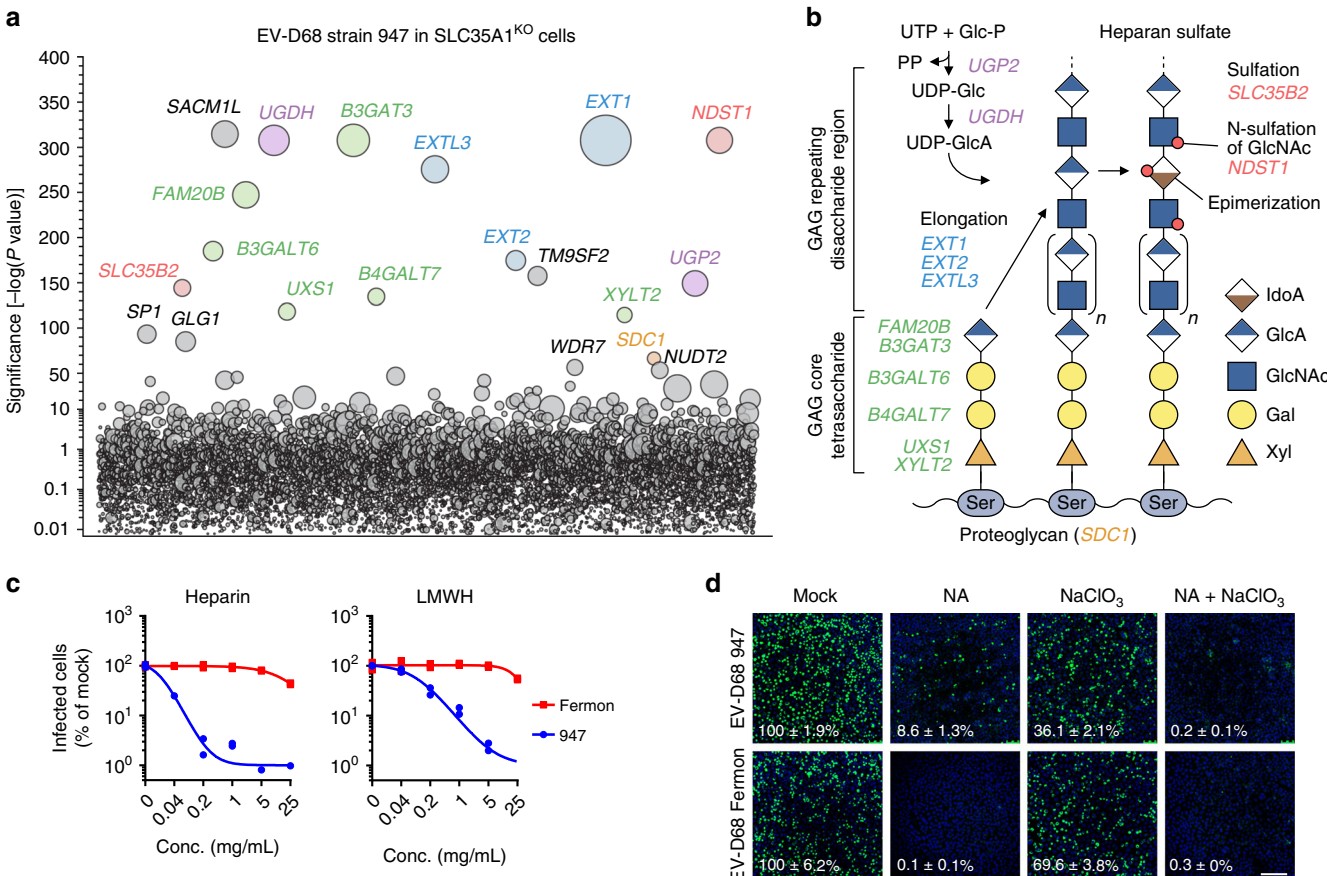

**Fig. 1** Both sialic acid and sulfated glycosaminoglycans can serve as EV-D68-947 receptors. **a** Haploid genetic screen for host factors of EV-D68-947. Each circle represents a gene, with size corresponding to the number of genetrap insertions per gene. The *y*-axis indicates the significance of enrichment of insertions in a gene, compared to an uninfected control population. Genes were randomly distributed on the *x*-axis. **b** Schematic overview of sulfated glycosaminoglycan (sGAG) synthesis, showing the functions of genes identified in the haploid screen, which are indicated in italics. UTP, uridine-5-triphosphate; UDP, uridine diphosphate; P, phosphate; Glc, glucose; GlcA, glucuronic acid; IdoA, iduronic acid; GlcNAc, N-acetylglucosamine; Gal, galactose; Xyl, xylose. **c** EV-D68 strains were incubated with various concentrations of heparin or low-molecular-weight-heparin (LMWH), followed by infection of HeLa-R19 cells, dsRNA staining and quantification of infected cells. Two technical replicates are shown. **d** HeLa-R19 cells were treated with neuraminidase (NA), sodium chlorate (NaClO₃) or a combination of both and infected with EV-D68, followed by staining of dsRNA (green) and nuclei (blue). Shown are representative confocal micrographs. Values denote the number of infected cells (mean ± s.e.m. of 3–4 technical replicates) as percentage of mock. Scale bar: 150 μm. The experiment was conducted three times with similar results

properties that eliminate the need for this host factor under certain conditions. EV-D68-947 originated from a patient with mild respiratory tract disease and was isolated by two passages in rhabdomyosarcoma (RD) cells[22]. Deep sequencing analysis revealed that two substitutions in capsid proteins VP2 and VP1 were introduced during isolation (Supplementary Fig. 2a). To validate this, we generated an infectious cDNA clone in which the original genome was restored and compared this virus (947 clinical) with EV-D68-947. Both genomes were able to replicate after transfection into T7 polymerase-expressing cells (Supplementary Fig. 2b), but only transfection of EV-D68-947, not of 947 clinical, yielded infectious virus that was able to disseminate and induce a cytopathic effect in cell lines (Supplementary Fig. 2c), supporting its adaptation to cultured cells. Although sGAG binding may be a consequence of this adaptation, the PLA2G16 independency of EV-D68-947 allowed us to use this virus as a tool to study the mechanism by which enteroviruses can circumvent PLA2G16 and to gain insights into its function during virus entry. In HeLa cells, EV-D68-947 required PLA2G16 only when using the Sia route (NaClO₃ treatment) but not when using the sGAG route (NA treatment) (Fig. 2b), whereas neither of these treatments affected PLA2G16 dependency of the Sia- and

sGAG-independent CV-B3 (Supplementary Fig. 2d). These data show that the reliance of EV-D68-947 on PLA2G16 is reduced by its ability to engage sGAGs as receptor. To confirm that the PLA2G16 independency of EV-D68-947 is due to its sGAG-binding capacity, we generated gain-of-function mutants of the strictly Sia-dependent EV-D68-2042, which differs from EV-D68-947 by only seven capsid residues[15] (Fig. 2c). Introduction of four surface-exposed capsid residues from EV-D68-947 into EV-D68-2042 (2042-4/5/6/7) enabled this virus to infect HAP1 CMAS^KO cells, another Sia-deficient cell line. Introducing a subset of these residues was tolerated for some combinations and led to reversion for other combinations. Mutant 2042-4/7, containing residues VP3 Glu^59 and VP1 Lys^271, could infect HAP1 CMAS^KO cells (Fig. 2c) and NA-treated HeLa cells (Fig. 2d and Supplementary Fig. 2e) and had gained sensitivity to neutralization with LMWH (Fig. 2d), showing that introduction of two residues from EV-D68-947 into EV-D68-2042 was sufficient to confer sGAG-binding capacity. Analysis of infectivity in PLA2G16^KO cells revealed that mutant 2042-4/7, when using sGAGs, was less reliant on PLA2G16 than was EV-D68-2042 (Fig. 2e and Supplementary Fig. 2f). This finding confirms that sGAG usage is associated with a reduced PLA2G16 requirement.

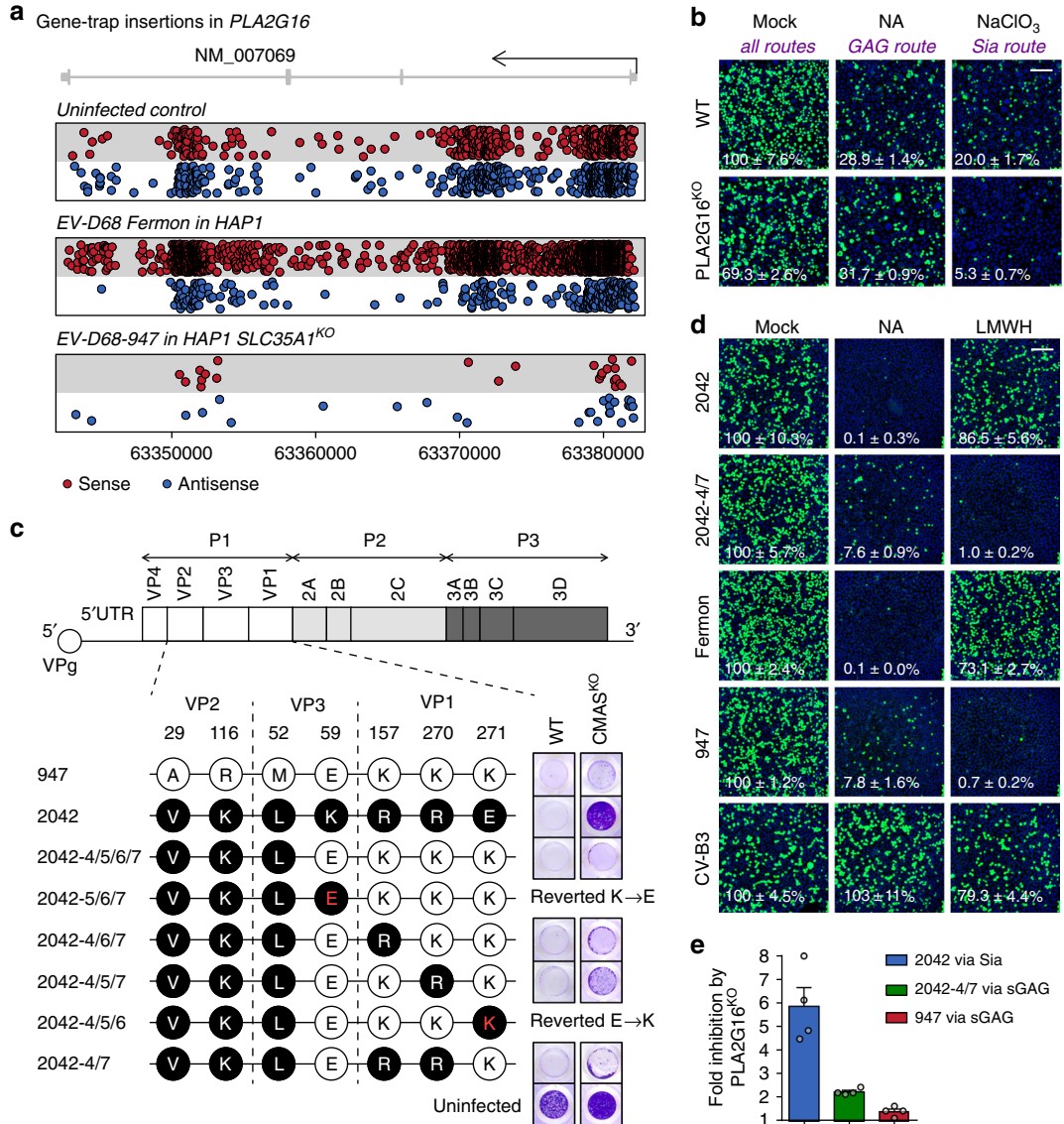

**Fig. 2** Engagement of sGAGs reduces the PLA2G16 dependency of EV-D68. **a** RefSeq gene structure of *PLA2G16* with genomic positions at the bottom. Each circle represents a sense (red) or antisense (blue) gene-trap insertion, a retroviral insert that disrupts the coding sequence of a gene when introduced in sense orientation into an intron, or in any orientation into an exon. An overall enrichment of disruptive sense gene-trap insertions (reflecting a role of *PLA2G16* in virus infection) was observed in a haploid genetic screen with EV-D68-Fermon[15], but not in a screen with EV-D68-947. **b** WT or PLA2G16[KO] H1-HeLa cells were treated with neuraminidase (NA), sodium chlorate (NaClO₃) or a combination of both and infected with EV-D68-947, followed by staining of dsRNA (green) and nuclei (blue). Values denote the number of infected cells (mean ± s.e.m. of four technical replicates) as percentage of mock-treated WT cells. Shown are representative confocal micrographs. The experiment was conducted twice with similar results. **c** Top, schematic representation of the enterovirus genome, with the 5′ untranslated region (5′UTR), precursor proteins P1-P3, capsid proteins VP1-VP4 and non-structural proteins 2A-2C, 3 A, 3B (or VPg), 3C, and 3D. Bottom, EV-D68-947 and 2042 differ in seven capsid residues, which were exchanged by site-directed mutagenesis. Red letters indicate positions at which mutants reverted back to the 947 genotype. WT or CMAS[KO] HAP1 cells were exposed to EV-D68 mutants and live cells were stained with crystal violet at 6 days post infection. **d** Viruses were incubated with low-molecular-weight-heparin (LMWH), followed by infection of mock- or NA-treated HeLa-R19 cells and staining of dsRNA (green) and nuclei (blue). Shown are representative confocal micrographs. Scale bars: 150 μm. Values denote the number of infected cells (mean ± s.e.m. of three technical replicates) as percentage of mock. **e** WT or PLA2G16[KO] H1-HeLa cells were infected and stained for dsRNA, followed by quantification of infected cells. Fold inhibition by PLA2G16[KO] was calculated by dividing the number of infected cells in PLA2G16[KO] cells by that in WT cells (see Supplementary Fig. 2f). Error bars: mean ± s.e.m. of four technical replicates. The experiment was conducted twice with similar results

**EV-D68-947 acid independency and ICAM-5 binding properties.** Although sGAG usage reduced the PLA2G16 dependency of 2042-4/7, this mutant was more dependent on PLA2G16 than EV-D68-947 (Fig. 2e), suggesting that additional factors influence the level of PLA2G16 requirement during virus entry. To explore whether EV-D68 strains require different uncoating cues, we compared sensitivity to the V-ATPase inhibitor Bafilomycin A1

(BafA1). Virus production of EV-D68 strains Fermon and 2042 was inhibited by BafA1, indicating that these strains, like the positive control RV-A2[5], rely on endosomal acidification during uncoating (Supplementary Fig. 3a). Similarly, analysis of the number of infected cells showed that the gain-of-function mutant 2042-4/7 was BafA1-sensitive (Fig. 3a), whereas BafA1 treatment hardly affected the negative control CV-B3 and EV-D68-947

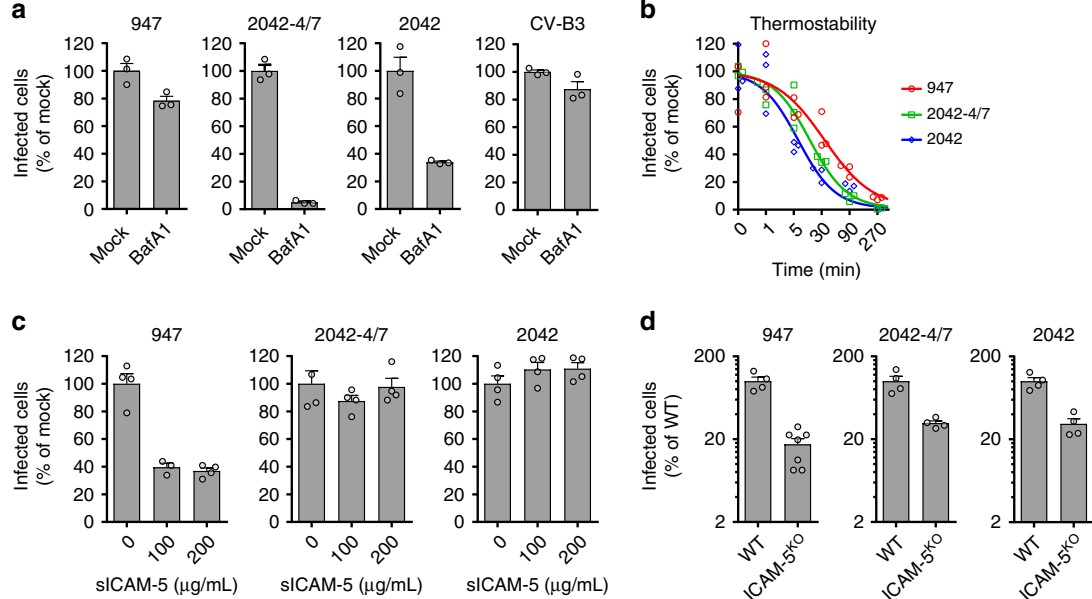

**Fig. 3** EV-D68-947 is acid-independent and has altered ICAM-5-binding properties. **a** HeLa-R19 cells were treated with 200 nM bafilomycin A1 (BafA1), infected and stained for dsRNA, followed by quantification of infected cells. The experiment was conducted twice with similar results. **b** Viruses were incubated at 48 °C for the indicated times, followed by infection of HeLa-R19 cells, dsRNA staining and quantification of infected cells. Four technical replicates with a fitted sigmoidal curve are shown. **c** Viruses were incubated with soluble ICAM-5 (sICAM-5), followed by infection of HeLa-R19 cells, dsRNA staining and quantification of infected cells. The experiment was conducted twice with similar results. **d** WT or ICAM-5$^{KO}$ HAP1 cells were infected and stained for dsRNA, followed by quantification of infected cells. Error bars represent the mean ± s.e.m. of three (**a**, **b**) or four (**c**, **d**) technical replicates

(Fig. 3a and Supplementary Fig. 3a). EV-D68-947 infection via the Sia route or the sGAG route was not inhibited by BafA1 treatment (Supplementary Fig. 3b), indicating that EV-D68-947 hardly requires endosomal acidification, but uncoats mainly via receptor-mediated destabilization. Thermostability assays showed that this acid independency was not due to lower particle stability (Fig. 3b). Therefore, we hypothesized that EV-D68-947 might be more extensively destabilized by the interaction with its receptor (s). Indeed, EV-D68-947 proved to be more sensitive than 2042-4/7 to neutralization with soluble ICAM-5 (Fig. 3c), but not with LMWH (Supplementary Fig. 3c), despite similar ICAM-5 requirements (Fig. 3d), indicating a stronger or more disruptive interaction between EV-D68-947 and ICAM-5. The conditions used in our neutralization experiment resemble those that were previously shown to stimulate A-particle formation and genome release in vitro[17,18]. Therefore, neutralization is likely due to receptor-mediated uncoating rather than competition between soluble ICAM-5 and cell-surface receptors. In summary, EV-D68-947 is acid-independent and has an enhanced or more disruptive interaction with ICAM-5. This virus is also less dependent on PLA2G16 when compared with the mutant 2042-4/7. These observations led to the hypothesis that PLA2G16 independency might be enabled by increased receptor-mediated virion destabilization.

**A basic patch in EV-D68-947 as putative sGAG binding site.** To investigate whether sGAGs bypass PLA2G16 by destabilizing the EV-D68-947 particles, we studied the effects of different glycan receptor analogues on virion structure by cryo-EM. Structures of EV-D68-947 were determined after incubation of the virus either in the absence of a receptor analogue, with 6′SLN (a sialylated trisaccharide), with dp6 (a heparin-derived hexasaccharide), or with LMWH at 33 °C (Supplementary Figs. 4 and 5 and Supplementary Table 1). A 2.4 Å resolution structure of the virus in complex with 6′SLN showed that EV-D68-947 binds to Sia in nearly the same way as does EV-D68-Fermon, despite some

differences in the amino acids near the receptor-binding site (Fig. 4a, b). We did not observe density corresponding to dp6 or LMWH in the EV-D68-947 structure, either because these ligands, in contrast to Sia, interact with the virus via a multitude of low-affinity and/or transient electrostatic interactions or because the bound ligands do not occupy all icosahedral symmetry related sites[23]. Nevertheless, the structure of the EV-D68-947 native virion showed a basic patch (formed by VP1 residues Lys$^{268}$, Lys$^{270}$, Lys$^{271}$, and Arg$^{272}$) that is not present in EV-D68-Fermon (Supplementary Fig. 6a). This basic patch presumably serves as a binding site for the negatively charged sGAGs and contains the positively charged Lys at VP1 position 271, which conferred a sGAG-binding capacity to mutant 2042-4/7 (Fig. 2c, Supplementary Fig. 6b). Together, these data point towards putative sGAG-binding residues in EV-D68-947 near the Sia-binding site.

**sGAG analogues stimulate EV-D68-947 uncoating.** Engagement of sGAGs, but not Sia, resulted in a reduced PLA2G16 dependency (Fig. 2b). Structural analyses showed that, like 6′SLN, both sGAG analogues induced movements of the VP1 GH loop into the VP1 hydrophobic pocket, causing displacement of the pocket factor (Fig. 4c), as previously observed[16]. This indicates that both Sia and sGAGs can prime the uncoating process. By contrast, determination of the ratio between the number of EV-D68-947 full and empty particles after 1 h incubation at 33 °C showed that the sGAG analogues dp6 and LMWH, but not 6′SLN, promoted genome release (Fig. 4d and Supplementary Fig. 7a–d). Structural comparison of LMWH-induced empty particles with empty particles formed in the absence of a ligand showed that LMWH caused an enlargement of the openings in the capsid around the two-fold symmetry axes that serve as putative sites of genome egress[24,25] (Fig. 4e and Supplementary Fig. 7e). Together, these data demonstrate that sGAGs, but not Sia, stimulate genome release from EV-D68-947, providing a

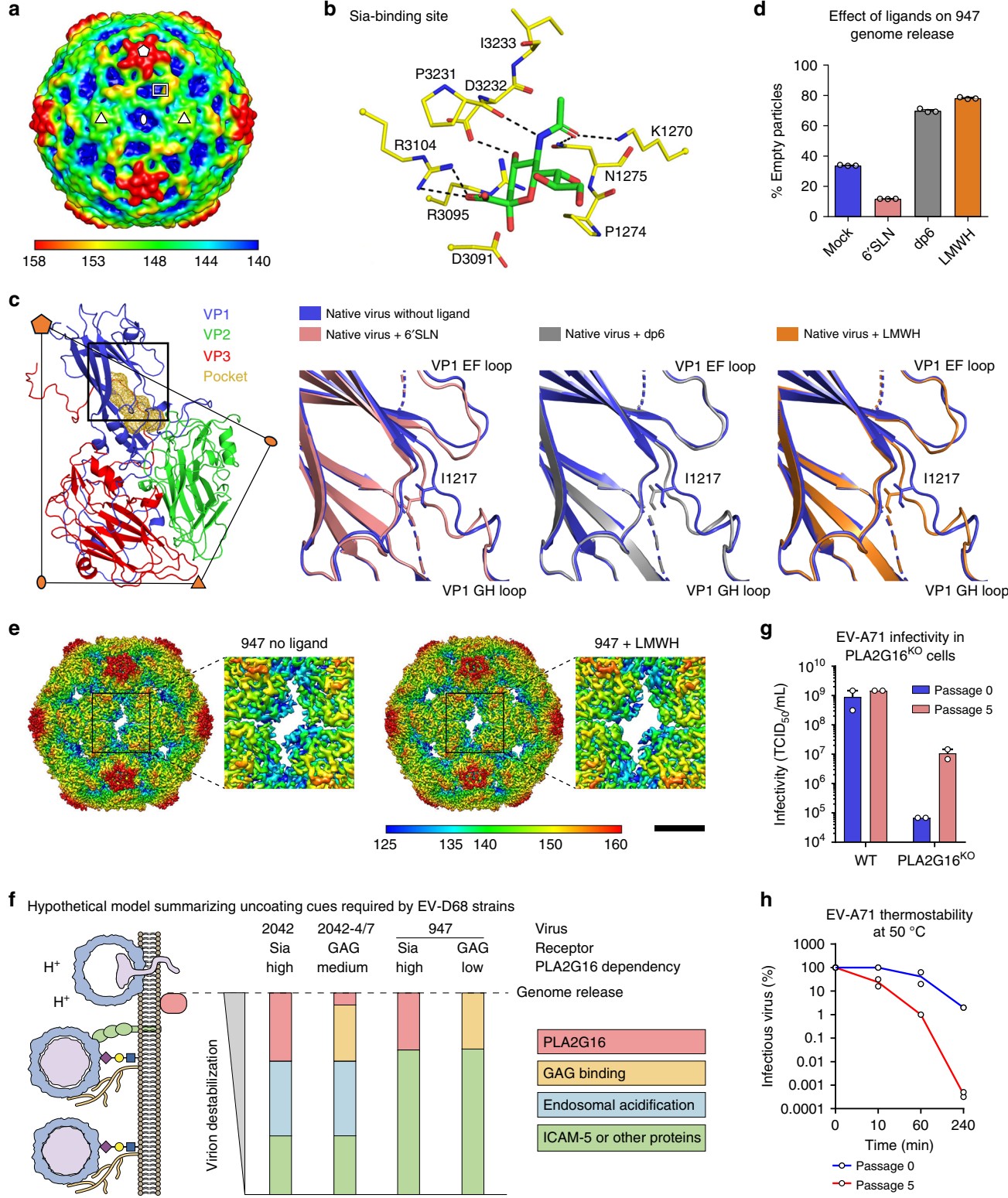

possible explanation for the capacity of EV-D68-947 to bypass PLA2G16 when infecting cells via sGAGs.

**PLA2G16 evasion via additional virion destabilization.** Our findings may be explained by a model in which subsequent uncoating cues orchestrate the priming of the EV-D68 particle for RNA release (Fig. 4f). The catalytic activity of PLA2G16 may constitute the ultimate event that triggers release of the genome.

Requiring such a host factor as a final checkpoint for RNA release might be beneficial for the virus, as it postpones genome release until the appropriate intracellular compartment has been reached. sGAGs binding, which is not strictly required for EV-D68-947 infection, may constitute an unnecessary destabilizing force that causes more extensive destabilization than is minimally required to prime the virion for PLA2G16-mediated infection. Such additional virion destabilization could disturb the delicate balance of uncoating cues and enable the virus to release its genome

**Fig. 4** sGAG analogues stimulate EV-D68-947 uncoating. **a** Structure of the EV-D68-947 native virion. Density map (low passed to 14 Å resolution) colored by radial distance to the particle center (Å). A pentangle, oval, and two triangles indicate five-, two-, and three-fold axes and delimit an icosahedral asymmetric unit. A rectangle outlines the limit of the close-up view in **b**. **b** Sialic acid-binding site in EV-D68-947. Shown are residues (yellow) within 4 Å from any atom of sialic acid (green). Black dashed lines indicate polar interactions. Residues are numbered by adding 1000 (VP1), 2000 (VP2), or 3000 (VP3) to their positions. **c** Receptor analogues induce conformational changes of EV-D68-947 to eject the pocket factor. Left panel: a biological capsid protomer is colored blue (VP1), green (VP2), red (VP3), and orange (the VP1 hydrophobic pocket). A rectangle delimits the three close-up views of the pocket (right panels): superimposed structures of native virus and virus with receptor analogues. VP1 Ile$^{217}$ moves such that it would clash with a pocket factor if it were present. **d** EV-D68-947 particles were incubated for 1 h at 33 °C with receptor analogues. Full and empty particles in cryo-EM images were counted. **e** Structure of EV-D68-947 empty capsids formed in the absence or presence of LMWH. VP2 residues 53–60, 98–100, and 240–242 and VP3 residues 148–150, which line the hole around the two-fold axis, are less ordered in emptied particles formed after LMWH treatment than in those formed without ligand. Color scale indicates radial distance to the particle center (Å). Scale bar: 35 Å. **f** Hypothetical model summarizing uncoating cues required by EV-D68 strains. Bar sizes are not based on quantitative measurements. **g** Infectivity of EV-A71 before (passage 0) and after (passage 5) five passages in H1-HeLa PLA2G16$^{KO}$ cells, as determined by end-point dilution on WT and PLA2G16$^{KO}$ H1-HeLa cells. TCID50: median tissue culture infective dose. Error bars represent mean ± s.e.m. of two biological replicates. **h** EV-A71 was incubated at 50 °C for the indicated times, after which the amount of intact virus was determined by end-point dilution on HeLa-R19 cells. Two biological replicates are shown

before encountering PLA2G16. Whether the alternative mode of endocytic uptake (clathrin-independent) via sGAGs (Supplementary Fig. 1c) is an additional factor influencing PLA2G16 (in) dependency, e.g. by altering the microenvironment in which uncoating takes place, is unknown. To investigate whether a link between virion stability and PLA2G16 dependency is unique for EV-D68 or also applies to other enteroviruses, we passaged EV-A71 (a member of the enterovirus A species) in PLA2G16-deficient knockout cells. This resulted in a mutant (passage 5) with multiple capsid substitutions (Supplementary table 2) that displayed a reduced PLA2G16 dependency (Fig. 4g). Concomitantly, this EV-A71 mutant was less thermostable than the parental virus (Fig. 4h), providing an additional line of support for a link between enterovirus stability and PLA2G16 dependency.

## Discussion

PLA2G16 was identified as an enterovirus host factor that facilitates the timely delivery of viral RNA into the cytoplasm before the virus is cleared by authophagy[8]. This enzyme is a promising target for broadly acting antiviral drugs, since it is required by all human enterovirus species tested thus far. We previously identified Sia as an EV-D68 receptor and showed that several strains, including EV-D68-947, can infect Sia-deficient cells[15]. In this study, we showed that EV-D68-947 is a dual receptor-binding virus that can use both sialylated glycans and sGAGs as receptors. Interestingly, we found that EV-D68-947 only requires PLA2G16 when infecting cells via Sia, but not via sGAGs. Consistently, introduction of a sGAG-binding site into the Sia- and PLA2G16-dependent strain EV-D68-2042 reduced its PLA2G16 dependency, although not to the same extent as EV-D68-947. We showed that this relative PLA2G16 independency of EV-D68-947 coincides with an enhanced sensitivity to neutralization by soluble ICAM-5, indicative of an enhanced or more disruptive interaction with this receptor. In addition, we showed that sGAGs stimulate EV-D68-947 genome release in vitro (Fig. 4d). While most enterovirus uncoating receptors are essential uncoating cues, sGAGs are not required for EV-D68-947 uncoating per se, but probably form an extra layer of destabilization that disturbs the natural balance of uncoating cues. Together, these results suggest that PLA2G16 independency is enabled by additional receptor-mediated virion destabilization. Finally, the observation that EV-A71 could adapt to replication in the absence of PLA2G16 via mutations that reduced virion stability (Fig. 4g, h) further implicates additional virion destabilization as a mechanism to bypass PLA2G16.

Structural analysis showed that Sia binds to EV-D68-947 in the same site as in EV-D68-Fermon and induces pocket factor

displacement. Incubation of EV-D68-947 with sGAG analogues not only caused pocket factor displacement but also promoted genome release, by enlarging the putative openings for genome egress at the two-fold axes via a yet unknown mechanism. Unfortunately, we were unable to determine the exact binding site of sGAGs on the EV-D68-947 particle by cryo-EM. Nevertheless, introducing two surface residues into EV-D68-2042 (Fig. 2c) conferred a sGAG-binding capacity and pointed towards putative sGAG-binding residues in a basic patch near the Sia-binding site, which is in a similar location as the sGAG-binding site in FMDV[26]. One of these putative GAG-binding residues, VP3 Glu$^{59}$, is present in most other published EV-D68 strains, whereas VP1 Lys$^{271}$ is absent in other clade B strains and rarely occurs in other clades[27]. Analysis of viral sequences in the clinical sample from which EV-D68-947 was isolated, revealed that the isolated virus had adapted to cultured cells via two capsid substitutions, including VP1 Lys$^{271}$. Hence, it remains to be established whether EV-D68-947 originally possessed a sGAG-binding capacity or acquired this property during cell passaging. Future research should determine whether sGAGs serve as EV-D68 receptors in vivo, by studying non-cell-adapted strains in physiologically relevant models like human airway epithelial (HAE) cultures.

In addition to glycan receptors, EV-D68 engages the cell-surface protein ICAM-5 as a receptor that promotes virus uncoating[17]. In this study, we confirmed the role of ICAM-5 as an EV-D68 receptor by showing that ICAM-5 knockout reduces infection by EV-D68 strains Fermon, 947, and 2042, even though this receptor was not essential for infection (Fig. 3d and Supplementary Fig. 1d). Although ICAM-5 knockout caused a similar reduction in infection for these strains, we found that soluble ICAM-5 has a different neutralizing effect on EV-D68-947 than on EV-D68-2042 and mutant 2042-4/7 (Fig. 3c). This may be due to an enhanced interaction with ICAM-5 or, alternatively, ICAM-5 may have a different destabilizing effect on EV-D68-947. Because the EV-D68 capsid residues directly involved in ICAM-5 binding are unknown, structural studies are needed to elucidate the details of this interaction. Such studies may also shed more light on the possible link between the neuron-specific protein ICAM-5 and neurological disease induced by different EV-D68 strains.

Together, our data reveal an unexpected connection between receptor binding and the requirement of PLA2G16 in cells, representing the first and last steps of enterovirus entry, respectively. This suggests that PLA2G16 plays a role in a virus-associated entry step, such as transmembrane pore formation or RNA translocation across the membrane. PLA2G16 might serve to initiate pore formation, improve pore permeability, or enhance

pore stability, thereby accelerating RNA release or steering the RNA toward a capsid opening that faces the endosomal membrane. PLA2G16 might exert its function directly, by interacting with viral proteins or RNA, or indirectly, by creating a lipid environment that is favorable for virus entry. The requirement of PLA2G16 as an uncoating cue might allow an enterovirus to recognize its destination, acting as the final checkpoint warranting the correct time and location of genome release. Additional virion destabilization could make this final checkpoint redundant, leading to premature genome release in the absence of PLA2G16. This implies that the development of resistance against inhibitors of this pan-enterovirus host factor is likely accompanied by premature virus uncoating and a fitness cost in vivo, which underscores the promise of PLA2G16 as a target for broad-range antiviral drugs.

## Methods

**Cells and viruses**. HAP1, HAP1 CMAS$^{KO}$, and HAP1 ICAM-5$^{KO}$ cells were obtained from Horizon Discovery Group plc (Cambridge, UK). HAP1 B3GALT6$^{KO}$, HAP1 SLC35A1$^{KO}$, H1-HeLa, and H1-HeLa PLA2G16$^{KO}$ cells were obtained from Thijn Brummelkamp (Netherlands Cancer institute, Amsterdam). HeLa-R19 cells were obtained from G. Belov (University of Maryland and Virginia-Maryland Regional College of Veterinary Medicine, US). Huh7/Lunet/T7 cells were obtained from Ralf Bartenschlager (Heidelberg University Hospital, Germany). Rhabdomyosarcoma (RD) cells were obtained from the European Collection of Cell Cultures (Catalogue No.: 85111502). HAP1 cells were cultured in Iscove's Modified Dulbecco's Medium (IMDM, Lonza) containing 10% (v/v) fetal calf serum (FCS). HeLa, Huh7/Lunet/T7 cells, HEK293T (ATCC CRL-3216) and RD cells were cultured in Dulbecco's Modified Eagle Medium (DMEM, Lonza) containing 10% (v/v) FCS. All cell lines tested negative for mycoplasma contamination. EV-D68-Fermon (CA62-1), EV-D68-947 (4310900947), and EV-D68-2042 (4310902042) were described previously[15] and, along with EV-A71 (BrCr strain), were obtained from the National Institute of Public Health and the Environment (RIVM), Bilthoven, The Netherlands. The genomic sequence of EV-D68-947 in the patient sample was determined by Baseclear B.V., Leiden, The Netherlands. RV-A2 was obtained from Joachim Seipelt, Medical University of Vienna, Austria. CV-B3 (Nancy strain) was obtained by transfecting in vitro-transcribed RNA derived from full length infectious clone p53CB3/T7[28].

**Chemicals and reagents**. The following chemicals and reagents were used in this study: *Arthrobacter ureafaciens* neuraminidase (Roche, 10269611001); heparin (Sigma-Aldrich, H4784); low-molecular-weight-heparin (Sigma-Aldrich, 1304118 USP); heparan sulfate (Iduron, GAG-HS01 BN1); hexasaccharide dp6 (Dextra Laboratories, ID1006); fondaparinux (Sigma-Aldrich, SML 1240); chlorpromazine (Sigma-Aldrich, C8138); sodium chlorate (Sigma-Aldrich, 1.06420 EMD Millipore); Recombinant human ICAM-5 (R&D systems, 1950-m5-050); bafilomycin A1 (Enzo Life Sciences, BML-CM110-0100); 6′SLN (6′-sialyl-*N*-acetyllactosamine, V-LABS, Inc.). Oligonucleotides used in this study are listed in Supplementary Table 3.

**Haploid genetic screen with EV-D68-947 in SLC35A1$^{KO}$ cells**. To generate gene-trap retrovirus, HEK293T cells were transfected with packaging plasmids and gene-trap plasmids[29]. Gene-trap retroviruses were harvested and used to mutagenize HAP1 SLC35A1$^{KO}$ cells. For selection with virus, 100 million mutagenized HAP1 SLC35A1$^{KO}$ cells were seeded in 14 T175 flasks, followed by infection with EV-D68-947. Resistant colonies were recovered after approximately two weeks, pooled and used for genomic DNA isolation and sequencing. Gene-trap insertion sites were amplified using a linear amplification -PCR protocol with a single biotinylated primer using 120 amplification cycles[30]. The resulting single-stranded DNA products were immobilized on M270 streptavidin-coated magnetic beads, washed and ligated to a single-stranded DNA linker, followed by exponential PCR amplification to introduce Illumina adapter sequences[30]. Finally, the PCR products were purified and subjected to Illumina sequencing. To map gene-trap insertion sites, sequence reads were aligned to the human genome (hg19) not filtering for close reads[31]. The number of disruptive gene-trap insertions (in sense orientation of the gene or mapping to exons) was compared to those present in a population of uninfected control cells, using a one-sided Fisher's exact test[31]. P values were false discovery rate (FDR)-corrected (Benjamani–Hochberg).

**Infectivity assays**. Cells were incubated with virus for 1 h at 37 °C, supplied with fresh medium and incubated at 37 °C for a total of 8 h, which represents a single replication cycle. In virus production assays, cells were subjected to three freeze/thaw cycles and virus titers were determined by end-point dilution. In neutralization assays, viruses were pretreated with ligands for 1 h at 37 °C. Neuraminidase treatments were performed by incubating cells with a 1:30 dilution of *A. urefaciens* NA in serum-free medium for 1 h at 37 °C. NaClO3 treatments were

performed by culturing cells in DMEM + 10% (v/v) FCS with 80 mM NaClO3 for at least five days to ensure depletion of sulfated glycans. To test whether viruses require endosomal acidification, cells were exposed to the indicated concentration of bafilomycin A1 for 1 h before infection, during infection and at least 5 h after infection. To block clathrin-mediated endocytosis, cells were exposed to 25 μM chlorpromazine for 30 min before infection, during infection and 3 h after infection.

**Immunofluorescence assays**. Cells were fixed by submersion in a 4% paraformaldehyde solution for 15 min. Fixed cells were stained with 1:2000 diluted mouse monoclonal anti-dsRNA (J2; English and Scientific Consulting) or 1:1000 diluted rabbit polyclonal antiserum against EV-D68 Fermon capsids (obtained from RIVM). Cells were examined by confocal microscopy (Leica SPE-II) and the number of infected cells was quantified with ImageJ.

**Construction of EV-D68 infectious cDNA clones**. Viral RNA was isolated from passage 3 (RD3) of strains 947 and 2042 (Genbank: 4310900947_RD3, KT231898; 4310902042_RD3, KT231902) and used to construct infectious cDNA clones. The 5′ and 3′ halves of the genomes were amplified separately and combined using an internal *SpeI* restriction site. Subsequently, the complete genomes were cloned into *XmaI* and *SalI* sites of pRib-CB3-Luc[32] from which the CB3-Luc region was deleted using the same enzymes, to yield pRib-EV68-947 and pRib-EV68-2042. Sequence analysis confirmed that the sequence of the viral cDNA was identical to that of the viral RNA. Mutant EV-D68-5/6/7 was made by inserting an *Eco81I-SpeI* fragment of EV-D68-947 into pRib-EV68-2042. Other mutants were generated by sequential introduction of single mutations into pRib-EV68-947 or pRib-EV-D68-5/6/7 using the Q5 site-directed mutagenesis kit (New England biolabs). Mutant viruses were generated by transfection of plasmids into Huh7/Lunet/T7 cells, which initiate transcription from the T7 promotor lying upstream of the viral cDNA. The resulting virus was passaged once in RD cells followed by verification of the genomic sequence.

**Virus growth and purification**. To generate EV-D68-947 for cryo-EM experiments, RD cells were infected with the virus using a multiplicity of infection of about 0.01. At about 60 h post infection, cells and supernatant were harvested and separated by centrifugation. The pellets were subjected to multiple cycles of freezing and thawing, followed by removal of cell debris via centrifugation. All supernatant was combined and spun down at 278,000 x g (Ti 50.2 rotor) for 2 h. The resultant pellets were resuspended in 250 mM HEPES, 250 mM NaCl, pH 7.5 (buffer 1), and sequentially treated with MgCl2, DNase, RNase, trypsin, EDTA solution (pH 9.5), and sodium n-lauryl-sarcosinate with a final concentration of 5 mM, 10 μg mL$^{-1}$, 0.25 mg mL$^{-1}$, 0.8 mg mL$^{-1}$, 15 mM, and 1% (w/v), respectively. The resultant sample was spun down, yielding pellets that were then resuspended in buffer 1. The crude virus was purified through a potassium tartrate density gradient (10–40% (w/v)) using a SW 41 Ti rotor. Purified virus was stored in phosphate buffer saline at 4 °C. All other virus stocks were produced by propagating virus in HeLa-R19 cells, subjecting cells and supernatant to three cycles of freezing and thawing, and centrifugation to remove cell debris.

**Cryo-electron microscopy**. For structure determination, purified EV-D68 947 was treated independently with 10 mg mL$^{-1}$ 6′SLN for 30 min (dataset virus-6′SLN), with 5 mg mL$^{-1}$ LMWH for 30 min (dataset virus-LMWH), with 5 mg mL$^{-1}$ dp6 for 1 h (dataset virus-dp6), or without a receptor analogue for 30 min (dataset virus-alone). These treatments were all at 33 °C. Aliquots of 2.7 μl of the resultant sample was applied onto copper grids with a continuous Lacey carbon film (400 mesh, Ted Pella). After blotting for about 7 s, the grid was plunge frozen into liquid ethane using a Cryoplunge 3 system (Gatan). Movies of EV-D68 particles embedded in a thin layer of vitreous ice were imaged on an FEI Titan Krios transmission electron microscope operated at 300 kV and equipped with a K2 Summit direct electron detector (Gatan). Data were automatically collected using the program Leginon[33]. The dose rate was about 4 e$^-$ pixel$^{-1}$ s$^{-1}$ for dataset virus-6′SLN and about 8 e$^-$ pixel$^{-1}$ s$^{-1}$ for all other datasets. Dataset virus-alone was collected at a nominal magnification of ×22500 in electron counting mode with defocus ranging from 0.5 to 3.1 μm. This gave a pixel size 1.30 Å pixel$^{-1}$ at the specimen level. A total electron dose of ~33 e$^-$ Å$^{-2}$ was fractionated into 28 frames with a frame rate of 250 ms frame$^{-1}$. Datasets virus-dp6 and virus-LMWH were collected at a nominal magnification in super resolution mode, giving a super resolution pixel size of 0.65 Å pixel$^{-1}$. The defocus range was 0.7–5.4 μm (dataset virus-dp6) and 0.6–4.5 μm (dataset virus-LMWH). A total electron dose of ~ 33 e$^-$ Å$^{-2}$ was fractionated into 35 frames with a frame rate of 200 ms frame$^{-1}$. Dataset virus-6′SLN was collected at a nominal magnification of ×29000 in super resolution mode with a defocus range of 0.4–3.6 μm. This gave a super resolution pixel size of 0.50 Å pixel$^{-1}$. A total electron dose of ~ 25 e$^-$ Å$^{-2}$ was fractionated into 62 frames with a frame rate of 100 ms frame$^{-1}$.

To test the effect of receptor analogues on viral stability, purified EV-D68 947 was incubated with 10 mg mL$^{-1}$ 6′SLN, 5 mg mL$^{-1}$ dp6, or 5 mg mL$^{-1}$ LMWH at 33 °C for 1 h. These complexes were formed at ligand:virus molar ratios of $1.6 \times 10^5$:1 (6′SLN), $2.9 \times 10^4$:1 (dp6) and $6.5 \times 10^3$:1 (LMWH). Considering the binding unit of sGAGs to be a disaccharide unit, the ligand:virus molar ratio was

8.6 × 10$^4$:1. As a control, virus was incubated in the absence of a receptor analogue. Cryo-EM sample preparation was performed as described above. Electron micrographs were manually collected at a nominal magnification of ×29500 using a total electron dose of ~25 e$^-$ Å$^{-2}$ on an FEI Phillips CM200 TEM operated at 200 kV and equipped with a 4k × 4k charged coupled device (Gatan). The defocus ranged from 0.7 to 6.0 μm. Particles were selected using e2boxer.py (part of the EMAN2 program package[34]), as will be detailed below. The number of full and empty particles was counted by three individuals. The total number of particles used for counting was 3690 (virus alone), 3023 (virus with dp6), 3107 (virus with LMWH), or 3492 (virus with 6′SLN).

**Image processing**. Movie frames were aligned using the program MotionCor2[35]. In this process, the first frame that had a large motion was discarded, and a reported dose weighting scheme was employed to down-weight the high resolution information in late frames[36]. The aligned frames were summed up and binned by a factor of two to yield individual electron micrographs. Binning was performed for all datasets except for dataset virus-alone at this stage. The non-dose-weighted micrographs were utilized to estimate contrast transfer function (CTF) parameters using the program CTFFIND4[37]. Projections derived from a three-dimensional (3D) reconstruction of EV-D68 strain US/MO14-18947[20] were low pass filtered to 40 Å resolution and served as templates for particle selection from the dose-weighted micrographs using e2boxer.py[34]. Particle images were then extracted and subjected to two-dimensional classification using the program Relion[38]. This procedure allowed for separation of full and empty particles and for removal of low-quality particles.

Samples of EV-D68 947 contained a mixed population of full particles after about 2 weeks of storage at 4 °C. For datasets virus-alone and virus-dp6, images of full particles (5.20 Å pixel$^{-1}$) were subjected to 3D classification using the program Relion, in which icosahedral symmetry was imposed[20]. This process enabled selection of a subset of particles that represented full virions rather than expanded uncoating intermediates (e.g., the A(altered)-particle). The following procedures were employed to reconstruct each of the six cryo-EM maps in this work using the program jspr[39]. These were full-native (dataset virus-alone), emptied (dataset virus-alone), full-dp6, full-LMWH, emptied-LMWH, and full-6′SLN. In brief, particle images were divided into two halves. For each half, particle images (5.20 Å pixel$^{-1}$) were used to generate multiple initial models by assigning random orientations to each particle. A suitable initial model was selected and used as an initial 3D reference for the refinement of particle orientations and centers. This procedure was extended to particle images with a pixel size of 2.60 Å pixel$^{-1}$ and subsequently to those with a pixel size of 1.30 Å pixel$^{-1}$. To improve the resolution of the reconstruction, parameters for particle orientations, centers, defocus, astigmatism, scale, beam tilt, and anisotropic magnification distortion[40,41] were then included in the refinement process. The resolution of the final cryo-EM map was estimated based on Fourier shell correlation (FSC) between two independently calculated half maps that were masked with a soft mask using an FSC cutoff of 0.143[42,43]. Maps were sharpened using a negative B-factor and low pass filtered using an FSC based filter[42]. In this process, the effect of modulation transfer function of the detector on the map was also corrected.

**Model building and refinement**. The following procedures were applied to each of the six structures presented in this work. A predicted atomic model of strain 947 was generated based on amino acid sequence comparison of strains 947 and US/MO14-18947 using SWISS-MODEL[44]. This model was refined against the cryo-EM map in real space using the program Phenix[45,46], followed by model rebuilding in Coot[47]. Then, a segment of the cryo-EM map was generated by selecting grid points within a radius of 5 Å from any atom of the current model. This map segment was back-transformed into structure factors, against which the B-factors, atomic positions, and occupancies of the atomic model was refined in reciprocal space using the program REFMAC5[48]. The resultant coordinates were further refined in real space using Phenix in which 60-fold non-crystallographic symmetry constraints were applied. Water molecules were added in Coot. The final atomic modes were assessed based on the criteria of MolProbity[49]. Figures were produced using Chimera[50], Pymol (https://pymol.org), and RIVEM[51].

**Reporting summary**. Further information on research design is available in the Nature Research Reporting Summary linked to this article.

## Data availability
The atomic coordinates of full-dp6, full-native, emptied, emptied-LMWH, full-LMWH, and full-6′SLN have been deposited in the Protein Data Bank with the accession codes 6CV1, 6CV2, 6CV3, 6CV4, 6CV5, and 6CVB. The cryo-EM reconstructions of full-dp6, full-native, emptied, emptied-LMWH, full-LMWH, and full-6′SLN have been deposited with the Electron Microscopy Data Bank under the accession codes EMD-7632, EMD-7633, EMD-7634, EMD-7635, EMD-7636, and EMD-7638. All relevant data are available from the authors. A reporting summary for this article is available as a Supplementary Information file. The source data underlying Figs. 1a, c, d, 2b, d, 3a–d, 4d, g, h and Supplementary Figs. 1a–d, 2d–f, 3a, c are provided as a Source Data file.

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

## Acknowledgements

Funding was provided by the Netherlands Organization for Scientific Research grant NWO-VICI-91812628 (to F.J.M.v.K.) and by National Institute of Health of the United States, NIAID grant AI011219 (to M.G.R.) We thank Jacqueline Staring for assistance with the genetic screen, Elmer Stickel and Vincent Blomen for assistance with data analysis, and prof. dr. Thijn Brummelkamp for his guidance. We thank Prof. Dr. Geert-Jan Boons for his valuable advice regarding the choice of sGAG analogs. We are grateful to Thomas Klose, Yingyuan Sun, Wen Jiang, and Valorie Bowman for help with cryo-EM analysis.

## Author contributions

J.B. was involved in performing the genetic screen. J.B., H.L., A.L.W.v.V., M.W. and R.W.R. performed infectivity assays. Y.L. performed cryo-EM experiments and data processing. A.L.W.v.V. and J.W.d.B. prepared mutant viruses. P.O. and A.M. analyzed viral sequences in patient material. J.B., H.J.T. and F.J.M.v.K designed the project. M.G.R and F.J.M.v.K. supervised and supported the project. J.B., Y.L., H.J.T and F.J.M.v.K. wrote the paper.

## Additional information

**Competing interests:** The authors declare no competing interests.

