## [Peer Review File · Nature Communications]

Reviewers' comments:

Reviewer #1 (Remarks to the Author):

General comments

Sialic acid (Sia) has been proposed as a receptor for EV-D68. This has been proved by previous work by the authors group. In the same study, they also found that some of the recently isolated EV-D68 strains were able to infect without using Sia. In this manuscript, the authors performed haploid genetic screen using Sia-KO cells to identify another receptor for Sia-independent pathway. They found a number of hits related to GAG biosynthesis. Furthermore, GAG analogs can inhibit viral infection and removal of sulfates on the cells reduced the infection efficiency. These results clearly showed that GAG can serve as another receptor for some strains of EV-D68 including EV-D68-947. The authors also noticed that PLA2G16 was not among the hits in the genetic screen, and investigated if PLA2G16 is not required in the GAG-mediated infection. They finally clarified that EV-D68-947 can infect the cells in a PLA2G16-independent and acid-independent manner. Structural analysis revealed that unlike Sia GAG is able to induce genome RNA release on EV-D68-947. The authors hypothesized that the evasion of PLA3G16 might be enable by an extensive receptor-mediated virion destabilization.

This study first identified GAG as another receptor for some EV-D68 strains, this pathway ia independent of pan-enterovirus host factor PLA2G16. These findings are very important and significantly contribute to elucidate mechanism of EV-D68 infection and picornavirus infection in general. The experiments were performed almost properly. However, this review has several concerns regarding to the interpretation of the results.

Specific comments

1. Lines 92-94. The authors described that virus strains with dual receptor emerged during the passages in RD cells. Does it mean GAG-dependent viruses are rare or not present in naturally circulating population?
2. The authors hypothesized that the evasion of PLA3G16 might be enable by an extensive receptor-mediated virion destabilization. Sia was able to expel the pocket factor from the virion (ref 12) but VP4 and genome RNA release was not observed. In contrast, GAG can initiate genome RNA release as shown in Fig 4d. However, this reviewer thinks extensive destabilization does not solely explain PLA2G16-independency. For example, PVR alone is able to induce release of VP4 and genome RNA when it is incubated with the virion, but PV infection is dependent on PLA2G16. Dependency may not be simply determined by receptor itself, it may be determined by interacton between the receptor and other unknown host factor(s). Therefore, this reviewer thinks that the title of the manuscript may not be adequate. The interpretation is largely based on success of infection in genetically modified cells. For example, ICAM-5 has been proposed as a receptor for EV-

D68 (ref14). However, this was not clearly introduced in introduction section. The relationship between the Sia and ICAM-5 and that between GAG and ICAM-5 has not been described well. In Fig 4c, the neutralization of 947 strain is shown. Is this neutralization occurred because of the competition of the sICAM-5 to ICAM-5 to cell surface or because of the in vitro uncoating before infection? In Fig. 4d, 947 infection looked dependent on ICAM-5. Are ICAM-5 KO cells used in this study or Sia/ICAM-5 dKO cells are used? If dKO cells are not used, the result is a mixture of Sia- and GAG-pathway.

3. It is also necessary to show when and where the uncoating occurred in GAG pathway. Microscopic studies like Fig 4 of ref 4 may be helpful to understand what is happening in the cells.
4. Fig 4f. How did the authors determine the ratio of each factor?
5. Line 116 and 486: f should be e.

Reviewer #2 (Remarks to the Author):

Comments to the manuscript “Dependency on pan-enterovirus host factor PLA2G16 is determined by receptor usage”

(1) Overall comments

The manuscript “Dependency on pan-enterovirus host factor PLA2G16 is determined by receptor usage” describes the dual glycan receptor usage of Enterovirus D68 (EV-D68). The present study has demonstrated that EV-D68 strain is capable of bypassing adipose-specific phospholipase A2 (PLA2G16) by employing sulfated glycosaminoglycans (sGAGs) as a receptor, which added a new scientific evidence to say that it might be the potential mechanism of PLA2G16 evasion of EV-D68. Methods used in the study were scientifically sound.

Considering the recent worldwide outbreaks of EV-D68 and its increasing epidemiological significance, I feel that the manuscript has a scientific merit and is suitable for publication in Nature Communications. However, there are several questions that need to be answered before publication.

(2) Major comments

Page 3, line 79.

In the study, EV-D68-947 was shown to possess unique receptor binding properties compared to EV-D68-2042. The haploid genetic screen for EV-D68-947 did not have hits with ICAM-5, and the strain was capable of infecting HAP1 ICAM-5KO cells, which suggested that EV-D68-947 might be relatively ICAM-5 independent. ICAM-5 is known as a viral receptor for EV-D68 in the central nervous system [1], which raises a question that if there are any groups of circulating EV-D68 strains that are less capable of causing neurological diseases. I'd like to know if authors have any data regarding the prevalence of such strains with differentiated ICAM-5 dependency, and if so, the epidemiological significance of those strains should be noted in the discussion section.

In addition, authors showed that the amino acid sequences of EV-D68-947 and EV-D68-2042, the ICAM-5 dependent strain, were different by seven residues in the capsid region, which suggested that those mutations could be potentially associated with ICAM-5 dependency of EV-D68. It might be better mention in the discussion section regarding the prevalence of such mutations, based on the phylogenetic analysis using the amino acid sequences that were reported in previous studies.

(3) Minor comments

Page 3, lines 88-89.

The receptor binding properties of EV-D68-947, especially for ICAM-5; the viral receptor in the central nervous system, was intensively discussed in the study. I'd be curious to know if the source of the strain, the patient presented any neurological disorders other than the mild upper respiratory tract symptoms that were already described in the manuscript.

Page 4, lines 97-100.

I'd like to know if the amino acid sequences of the strain EV-D68-947 was identical to those of EV-D68-2042 in the regions other than structural proteins, VP1-3.

(4) References

[1] Wei, W. et al. ICAM-5/Telencephalin Is a Functional Entry Receptor for Enterovirus D68. *Cell Host Microbe* 20, 631–641 (2016).

Reviewer 3 (Remarks to the Author):

The manuscript by Baggen et al. describes a novel entry mechanism mediated by sulfated glycosaminoglycans (sGAGs) for enteroviruses which bypasses an established PLA2G16/SIA mediated

mechanism. They employ haploid screening to identify proteins potentially linked to the new mechanism, and KO cell lines to demonstrate independence of enterovirus EV-D68-947 from PLA2G16. The authors also employ chemical means to remove cellular glycan receptors and competitive binding with analogs to further demonstrate the reliance of the virus on these known or haploid screen predicted glycan receptors. Mutagenesis of a closely related virus (EVD68-2042) without these glycan binding properties identified the residues mediating the sGAG binding. ATPase inhibitors were used to show that EV-D68-947 did not require acidification during endocytosis to escape the endosome. Finally, single particle cryo-electron microscopy was used to visualize conformational changes in the capsid related to glycan binding and demonstrated similar rearrangements with different receptors. This manuscript puts forth a convincing case for a novel entry mechanism for this enterovirus, EV-D68-947 that would be of interest to readers of Nature Communications.

Minor concerns are listed below. Minor concerns: The title of the paper is misleading. While it is true the data demonstrates the new glycan receptor bypasses the PLA2G16, the PLA2G16 is not the major theme to the manuscript. Suggest revising to focus on the novel entry mechanism described.

The introduction should include more enterovirus background. This dearth is most apparent when later in the when CV-B3 and RV-A2 are mentioned. To the reader not familiar with enteroviruses, their use as controls makes no sense. Thus an introduction to the family that mentions these viruses would be informative.

The results were concise and convincing, however, suggest correcting/revising issues below: 1. Sulfate is spelled as both "sulfate" (e.g. results line70) and "sulphate" (supplemental figure 1). 2. Please write out words before using abbreviations such as RD cells on line 92. 3. Please introduce CV-B3 and RV-A2 as mentioned above. To a reader no fully familiar with enteroviruses, their choice use as controls is not obvious. 4. The basic patch proposed for sGAG binding is a proposal because the glycans were not ordered. This presentation suggests a definitive binding site without convincing mutagenesis data or density from single particle cryo-electron microscopy. Suggest using wording that ensures that the reader does not get the impression that this is the proven binding pocket for sGAGs.

The methods were understandable and detailed. A few items to clarify:

1. There is a reference to a "Nancy" on line 198 (penultimate line) in the cells and viruses section. Please clarify what this is in reference to.

2. Suggest providing the concentration of the ligands used for complexing for the Cryo-EM studies in the same units (mg/mL or mM) instead of both. Given the lack of density for the dp6 and LMWH in the complex maps, please provide information on the virus –glycan molar ratio in the complexes. Include a discussion of the possibility of differences in affinity for Sia and sGAGs.

3. Line 291, "expect" should be "except"

Figures were mostly appropriate. Minor corrections/clarifications needed:

1. In supplemental figure 1 revise “heparin sulphate” to heparin. Clarify what TCID50 is.
2. In figure 2 panel a, to readers not familiar with haploid screens or gene traps this needs more context. In addition, panel c refers to “Grey circles”, there are no grey circles in the actual figure, only black and white. Please correct or clarify.
3. In figure 4 panels b and c, the naming convention used for residues is readily obvious. Please clarify in the legend.

Response to reviewers manuscript of Baggen et al. entitled “Evasion of pan-enterovirus host factor PLA2G16 via excessive virion destabilization”

Previous manuscript title: “Dependency on pan-enterovirus host factor PLA2G16 is determined by receptor usage”.

First, we would like to thank the reviewers for their time to evaluate our submission and we thank them for their positive judgment. Our answers to the points raised by each of the reviewers are given below.

Reviewer #1:

General comments

Sialic acid (Sia) has been proposed as a receptor for EV-D68. This has been proved by previous work by the authors group. In the same study, they also found that some of the recently isolated EV-D68 strains were able to infect without using Sia. In this manuscript, the authors performed haploid genetic screen using Sia-KO cells to identify another receptor for Sia-independent pathway. They found a number of hits related to GAG biosynthesis. Furthermore, GAG analogs can inhibit viral infection and removal of sulfates on the cells reduced the infection efficiency. These results clearly showed that GAG can serve as another receptor for some strains of EV-D68 including EV-D68-947. The authors also noticed that PLA2G16 was not among the hits in the genetic screen, and investigated if PLA2G16 is not required in the GAG-mediated infection. They finally clarified that EV-D68-947 can infect the cells in a PLA2G16-independent and acid-independent manner. Structural analysis revealed that, unlike Sia, GAG is able to induce genome RNA release on EV-D68-947. The authors hypothesized that the evasion of PLA3G16 might be enable by an extensive receptor-mediated virion destabilization. This study first identified GAG as another receptor for some EV-D68 strains, this pathway ia independent of pan-enterovirus host factor PLA2G16. These findings are very important and significantly contribute to elucidate mechanism of EV-D68 infection and picornavirus infection in general. The experiments were performed almost properly. However, this review has several concerns regarding to the interpretation of the results.

Specific comments

1) Lines 92-94. The authors described that virus strains with dual receptor emerged during the passages in RD cells. Does it mean GAG-dependent viruses are rare or not present in naturally circulating population?

Our observation that VP1 Lys²⁷¹, one of the residues enabling sGAG usage (**Figure 2d**), was not present in the original EV-D68-947 patient sample, strongly suggests that this particular strain acquired the capacity to employ sGAGs as a consequence of cell culture adaptation.

In this revised version of the manuscript, we further confirmed this adaptation by restoring the 947 genome to its original sequence and comparing the infectivity of the resultant virus “947 clinical” with cell-adapted EV-D68-947. Although 947 clinical was able to replicate its genome following transfection into T7 polymerase-expressing cells

(**Supplementary Figure 2b**), it was unable to form infectious virus that could disseminate and induce cytopathic effect in several cell lines (**Supplementary Figure 2c**). This strongly supports the idea that EV-D68-947 adapted to cultured cells.

Nevertheless, based on these data concerning a single strain, we cannot exclude the possibility that other sGAG-binding strains circulate *in vivo*. Therefore, future investigations comparing many EV-D68 isolates in more physiologically relevant models like human airway epithelial cultures (HAE) should establish whether Sia or sGAGs serve as receptors *in vivo*. We have briefly mentioned this topic in the discussion section of the revised manuscript (lines 238-242).

2) *The authors hypothesized that the evasion of PLA2G16 might be enable by an extensive receptor-mediated virion destabilization. Sia was able to expel the pocket factor from the virion (ref 12) but VP4 and genome RNA release was not observed. In contrast, GAG can initiate genome RNA release as shown in Fig 4d. However, this reviewer thinks extensive destabilization does not solely explain PLA2G16-independency. For example, PVR alone is able to induce release of VP4 and genome RNA when it is incubated with the virion, but PV infection is dependent on PLA2G16. Dependency may not be simply determined by receptor itself, it may be determined by interacton between the receptor and other unknown host factor(s). Therefore, this reviewer thinks that the title of the manuscript may not be adequate. The interpretation is largely based on success of infection in genetically modified cells.*

As the reviewer correctly points out, receptor-mediated destabilization is not the only possible mechanism of PLA2G16 evasion. As summarized in Figure 4f, we propose that the requirement of PLA2G16 is determined by the cumulative activity of different uncoating cues, which together overcome virion stability.

In this revised version of the manuscript, we have added data showing that passaging of enterovirus A71 in PLA2G16 knockout cells yielded a PLA2G16-independent mutant with reduced thermostability (**Figures 4 g and h**). These data further confirm that there is a link between virion stability and PLA2G16 dependency.

For the above reasons, we have changed the title of the manuscript to: “Evasion of pan-enterovirus host factor PLA2G16 via excessive virion destabilization”. In addition, to clarify that receptor-mediated destabilization is not necessarily associated with PLA2G16 independency, we have removed the word “receptor-mediated” from the abstract.

3) *For example, ICAM-5 has been proposed as a receptor for EV-D68 (ref14). However, this was not clearly introduced in introduction section. The relationship between the Sia and ICAM-5 and that between GAG and ICAM-5 has not been described well.*

As requested, we have added information about this topic in the introduction section (lines 58-61).

4) In Fig 4c, the neutralization of 947 strain is shown. Is this neutralization occurred because of the competition of the sICAM-5 to ICAM-5 to cell surface or because of the *in vitro* uncoating before infection?

Indeed, the neutralization of EV-D68-947 by soluble ICAM-5 protein can be due to competition with cell surface receptors, due to ICAM-5-induced virus uncoating, or a combination of both mechanisms.

In a recent publication (Zheng, Q. et al. Atomic structures of enterovirus D68 in complex with two monoclonal antibodies define distinct mechanisms of viral neutralization. *Nat. Microbiol.* (2019).) it was shown that exposure of the EV-D68 genome is stimulated when the virus is incubated *in vitro* with ICAM-5 under the same conditions as in this study (37 °C in PBS; same protein supplier) and an ICAM-5 concentration (60 µg/ml) slightly below that used in this study. Thus, it is likely that the neutralization of EV-D68-947 shown in Figure 4c is a consequence of ICAM-5-mediated *in vitro* uncoating. We have addressed this topic in the results section of the revised manuscript (lines 153-156).

5) In Fig. 3d, 947 infection looked dependent on ICAM-5. Are ICAM-5 KO cells used in this study or Sia/ICAM-5 dKO cells are used? If dKO cells are not used, the result is a mixture of Sia- and GAG-pathway.

The experiments in Figure 3d and Supplementary Figure 1c were performed in ICAM-5 single knockout cells. Therefore, as the reviewer correctly points out, the virus can employ either Sia or GAGs to infect these cells. This observation also shows that no redundancy exists between ICAM-5 on one hand and Sia/GAG on the other hand, suggesting that ICAM-5 might play a distinct role (e.g. endocytosis) from the two glycan receptors.

6) It is also necessary to show when and where the uncoating occurred in GAG pathway. Microscopic studies like Fig 4 of ref 4 may be helpful to understand what is happening in the cells.

As suggested by the reviewer, the possibility exists that infection via sGAGs is associated with a different localization of virus uncoating. We addressed this question by investigating whether the route of endocytic uptake of EV-D68-947 differs between the Sia/sGAG infection routes, using the clathrin-mediated endocytosis (CME) inhibitor chlorpromazine. As shown in **Supplementary Figure 1c**, chlorpromazine inhibited infection of the strictly Sia-dependent EV-D68 strain 2042, indicating that this strain requires CME for internalization. In contrast, EV-D68-947 required CME only when infecting cells via Sia (NaClO₃-treated cells), but not when infecting cells via sGAGs (NA-treated cells). Thus, virus uptake via sGAGs differs from uptake via Sia and either does not require endocytosis or is mediated by a clathrin-independent pathway.

This observation implies that the site of uncoating may differ between the Sia/sGAG routes. Thus, it cannot be excluded that the alternative mode of uptake via sGAGs contributes to the evasion of PLA2G16, in addition to the direct destabilizing effect of sGAG binding on the virion. We therefore addressed this topic in the results section of the revised manuscript (lines 201-203).

It would be interesting to further elucidate the precise localization of virus uncoating via microscopic studies (as in Figure 4d of reference 4) and we have therefore discussed this possibility with our collaborators Prof. Thijn

Brummelkamp and Dr. Jacqueline Staring, who previously performed these experiments. They indicated that the microscopy data in Figure 4 were the result of several years of optimization and pointed out that it will be time-consuming to expand these assays with different conditions and to adapt them to a different virus strain. To obtain information about the subcellular location of uncoating, we will have to expand the technology described in Figure 4d of reference 4, by monitoring not only pore formation (GFP-Galectin-8) and viral genomes (RNA probes), but also tracking markers of various endocytic compartments. Furthermore, visualizing transmembrane pores with GFP-tagged galectin-8 requires using a very high multiplicity of infection, which is more challenging for EV-D68-947 (>100-fold lower titers) than for poliovirus, especially since the infectivity will be further reduced by NA and NaClO₃ pretreatment. Moreover, to detect the viral genome after its release from endosomes, a set of RNA probes specific for EV-D68-947 needs to be designed and optimized (for some viruses, probe sets did not work in the past). Due to these substantial practical challenges, we think that more profound investigation of the uncoating sites should be the subject of a follow-up study.

7) *Fig 4f. How did the authors determine the ratio of each factor?*

In the hypothetical model presented in Figure 4f, the exact size of each bar is arbitrary. Nevertheless, there are some constraints to their relative sizes: the “ICAM-5 or other proteins” bar for EV-D68-947 must be larger than the “endosomal acidification” bar for the other two strains, in order to explain the acid-independent infection by EV-D68-947. In addition, for EV-D68-947, the “GAG binding” bar must be similar in size to the “PLA2G16” bar, to explain why GAG binding omits the need for PLA2G16. To make sure the reader knows that these bars should not be interpreted as quantitative results, we have indicated this in the legend of Figure 4f.

8) *Line 116 and 486: f should be e.*

This change has been introduced into the manuscript text.

Reviewer #2:

Overall comments

The manuscript “Dependency on pan-enterovirus host factor PLA2G16 is determined by receptor usage” describes the dual glycan receptor usage of Enterovirus D68 (EV-D68). The present study has demonstrated that EV-D68 strain is capable of bypassing adipose-specific phospholipase A2 (PLA2G16) by employing sulfated glycosaminoglycans (sGAGs) as a receptor, which added a new scientific evidence to say that it might be the potential mechanism of PLA2G16 evasion of EV-D68. Methods used in the study were scientifically sound.

Considering the recent worldwide outbreaks of EV-D68 and its increasing epidemiological significance, I feel that the manuscript has a scientific merit and is suitable for publication in Nature Communications. However, there are several questions that need to be answered before publication.

Major comments

- 1) *Page 3, line 79: In the study, EV-D68-947 was shown to possess unique receptor binding properties compared to EV-D68-2042. The haploid genetic screen for EV-D68-947 did not have hits with ICAM-5, and the strain was capable of infecting HAPI ICAM-5KO cells, which suggested that EV-D68-947 might be relatively ICAM-5 independent. ICAM-5 is known as a viral receptor for EV-D68 in the central nervous system [Wei, W. et al. 2016], which raises a question that if there are any groups of circulating EV-D68 strains that are less capable of causing neurological diseases. I'd like to know if authors have any data regarding the prevalence of such strains with differentiated ICAM-5 dependency, and if so, the epidemiological significance of those strains should be noted in the discussion section.*

Wei W. *et al.* showed that ICAM-5 silencing inhibits EV-D68 infection of HEK293T cells in a single replication cycle by 4-5-fold (Figure S1b). Although ICAM-5 did not appear as a hit in the haploid screen, we have been able to confirm this role of ICAM-5 in EV-D68 infection and observed comparable ICAM-5 dependencies as described by Wei W. *et al.*, with similar ICAM-5 dependencies for strains Fermon, 2042, and 947 (**Figure 3d and Supplementary Figure 1d**).

Indeed, it will be interesting to know which residues control ICAM-5 binding, to compare their prevalence among clinical isolates and to find a possible link with neurological disease. However, to perform such analyses, a structure of EV-D68 in complex with ICAM-5 needs to be determined first. We have therefore mentioned this future research endeavor in the discussion section (lines 251-254).

- 2) *In addition, authors showed that the amino acid sequences of EV-D68-947 and EV-D68-2042, the ICAM-5 dependent strain, were different by seven residues in the capsid region, which suggested that those mutations could be potentially associated with ICAM-5 dependency of EV-D68. It might be better mention in the*

discussion section regarding the prevalence of such mutations, based on the phylogenetic analysis using the amino acid sequences that were reported in previous studies.

When comparing strains EV-D68-947 and EV-D68-2042, we did not observe a difference in their dependencies on ICAM-5 for infection of cells (**Figure 3d and Supplementary Figure 1d**) but found differences in their sGAG-binding capacities (**Figure 2d**) and their sensitivities to neutralization by ICAM-5 (**Figure 3c**). Indeed, these differences are probably caused by the seven amino acid variations shown in Figure 2c.

To evaluate the prevalence of these residues, we have compared the amino acid sequences of all published EV-D68 strains (see below). This revealed that the residues in EV-D68-947 that resulted from cell culture adaptation (VP2 Arg¹¹⁶ and VP1 Lys²⁷¹) are highly uncommon, whereas the other five residues frequently occur in other strains. In contrast, the corresponding five residues in EV-D68-2042 are highly uncommon. We have mentioned the prevalence of residues specific for EV-D68-947 in the discussion section (lines 235-236).

Prevalence of EV-D68-947 residues among published EV-D68 strains:

VP2 Ala²⁹: present in almost all published strains

VP2 Arg¹¹⁶: not present in any other strain

VP3 Met⁵²: present in almost all published strains

VP3 Glu⁵⁹: present in almost all published strains

VP1 Lys¹⁵⁷: other strains frequently have either Lys or Glu

VP1 Lys²⁷⁰: present in almost all published strains

VP1 Lys²⁷¹: Zhang, Scheuermann et al. 2016: Lys²⁷¹ does not occur in clade B1 strains, sometimes occurs in other clades. Glu is the most common residue at this position.

Minor comments

1) Page 3, lines 88-89. *The receptor binding properties of EV-D68-947, especially for ICAM-5; the viral receptor in the central nervous system, was intensively discussed in the study. I'd be curious to know if the source of the strain, the patient presented any neurological disorders other than the mild upper respiratory tract symptoms that were already described in the manuscript.*

No, the patient from which EV-D68-947 was originally isolated did not display any neurological symptoms, but suffered only mild respiratory disease, as is mentioned in the results section. (line 111).

2) Page 4, lines 97-100. *I'd like to know if the amino acid sequences of the strain EV-D68-947 was identical to those of EV-D68-2042 in the regions other than structural proteins, VP1-3.*

In addition to the seven capsid residues that differ between strains 947 and 2042, there are also three amino acid differences between these strains in the non-structural region. These are located in the proteins 3A, 3C, and 3D.

Reviewer #3:

The manuscript by Baggen et al. describes a novel entry mechanism mediated by sulfated glycosaminoglycans (sGAGs) for enteroviruses which bypasses an established PLA2G16/SIA mediated mechanism. They employ haploid screening to identify proteins potentially linked to the new mechanism, and KO cell lines to demonstrate independence of enterovirus EV-D68-947 from PLA2G16. The authors also employ chemical means to remove cellular glycan receptors and competitive binding with analogs to further demonstrate the reliance of the virus on these known or haploid screen predicted glycan receptors. Mutagenesis of a closely related virus (EVD68-2042) without these glycan binding properties identified the residues mediating the sGAG binding. ATPase inhibitors were used to show that EV-D68-947 did not require acidification during endocytosis to escape the endosome. Finally, single particle cryo-electron microscopy was used to visualize conformational changes in the capsid related to glycan binding and demonstrated similar rearrangements with different receptors. This manuscript puts forth a convincing case for a novel entry mechanism for this enterovirus, EV-D68-947 that would be of interest to readers of Nature Communications. Minor concerns are listed below.

Minor concerns:

- 1) The title of the paper is misleading. While it is true the data demonstrates the new glycan receptor bypasses the PLA2G16, the PLA2G16 is not the major theme to the manuscript. Suggest revising to focus on the novel entry mechanism described.***

Although we identified sGAGs as a novel receptor for EV-D68-947, our efforts to further characterize the sGAG-mediated entry mechanism were mainly intended to explain how this virus can circumvent the need for PLA2G16. Together, our findings converge into the conclusion that an imbalance in the cellular uncoating cues can make PLA2G16 redundant. In addition, we have now included additional data that reveal a link between EV-A71 stability and PLA2G16 dependency, which further broadens our message.

In addition, we prefer not to emphasize the sGAG-mediated entry mechanism in the title, to discourage readers to think that sGAGs serve as EV-D68 receptors *in vivo*, which should first be established via experiments with non-cell culture-adapted isolates in clinically relevant tissue models.

Although we understand the reviewer's concerns, we prefer that the title focuses on the link between PLA2G16 and virion destabilization, whereas the secondary messages about receptor usage can be found in the abstract.

- 2) The introduction should include more enterovirus background. This dearth is most apparent when later in the when CV-B3 and RV-A2 are mentioned. To the reader not familiar with enteroviruses, their use as controls makes no sense. Thus an introduction to the family that mentions these viruses would be informative.***

As the reviewer suggested, we have added more information about enteroviruses in the introduction section and have included background information about the uncoating mechanisms of the control viruses CV-B3, RV-A2, and EV-A71 (lines 31-40).

3) *The results were concise and convincing, however, suggest correcting/revising issues below:*

- a) *Sulfate is spelled as both “sulfate” (e.g. results line 70) and “sulphate” (supplemental figure 1).*
- b) *Please write out words before using abbreviations such as RD cells on line 92.*
- c) *Please introduce CV-B3 and RV-A2 as mentioned above. To a reader not fully familiar with enteroviruses, their choice use as controls is not obvious.*

We have introduced the changes suggested above into the revised manuscript.

- d) *The basic patch proposed for sGAG binding is a proposal because the glycans were not ordered. This presentation suggests a definitive binding site without convincing mutagenesis data or density from single particle cryo-electron microscopy. Suggest using wording that ensures that the reader does not get the impression that this is the proven binding pocket for sGAGs.*

As suggested, we have modified the sentences about this basic patch by repeatedly mentioning “putative” sGAG-binding residues (lines 162 and 177) and by removing the statement that EV-D68-947 has “adjacent receptor binding sites” (line 181), such that it should be clear to the reader that the location of sGAG binding is not definitively proven. Moreover, we added a more elaborate discussion of these putative sGAG-binding residues in the discussion section (231-236).

4) *The methods were understandable and detailed. A few items to clarify:*

- a) *There is a reference to a “Nancy” on line 198 (penultimate line) in the cells and viruses section. Please clarify what this is in reference to.*

We have indicated that “Nancy” is the name of a CV-B3 strain.

- b) *Suggest providing the concentration of the ligands used for complexing for the Cryo-EM studies in the same units (mg/mL or mM) instead of both. Given the lack of density for the dp6 and LMWH in the complex maps, please provide information on the virus –glycan molar ratio in the complexes.*

We have now indicated all ligand concentrations in the same unit (mg mL^{-1}) and have mentioned the virus:glycan molar ratios of the complexes in the methods section (lines 370-372).

Include a discussion of the possibility of differences in affinity for Sia and sGAGs.

We think that the interaction of the virus with sGAGs is different from the interaction with Sia, since sGAGs probably bind via a multitude of low-affinity interactions, as is now mentioned in the results section (lines 170-171). However, we cannot directly compare the binding affinity between sGAGs and 6'SLN, for the following reasons:

- 1) Although it is possible that a sGAG disaccharide unit has a lower affinity than 6'SLN, when calculating the binding affinity of polymeric sGAGs to the virus, multivalency needs to be taken into account, which is not the case for 6'SLN.
- 2) Different kinetic models may be required to describe the binding of 6'SLN and sGAGs. For instance, binding of sGAGs to the virus might involve more than one binding site characterized by different dissociation constants.

c) Line 291, “expect” should be “except”

This has been corrected.

5) Figures were mostly appropriate. Minor corrections/clarifications needed:

a) In supplemental figure 1 revise “heparin sulphate” to heparin. Clarify what TCID50 is.

The ligand “heparan sulfate” in Supplementary Figure 1a is not the same reagent as the “heparin” used in Figure 1c. We included heparan sulfate because this sGAG analogue is structurally more representative for cell-surface sGAGs than heparin, since it is more heterogeneous and has a lower degree of sulfation.

We have clarified “TCID50” in the legends of Supplementary Figures 1b and 3a and Figure 4g.

b) In figure 2 panel a, to readers not familiar with haploid screens or gene traps this needs more context.

As suggested, we have more elaborately explained the gene-trap insertions shown in the legend of Figure 2a.

In addition, panel c refers to “Grey circles”, there are no grey circles in the actual figure, only black and white. Please correct or clarify.

Indeed, the grey circles in Figure 2c were previously removed and replaced by red letters. This error has been corrected in the legend.

c) In figure 4 panels b and c, the naming convention used for residues is readily obvious. Please clarify in the legend.

We have clarified the residue numbering system in the legend of Figure 4.

The additional experiments included in this revised version of the manuscript were partly performed by a member of our lab that was not previously listed as a co-author. We have added her name (Heyrhyoung Lyoo) to the author list and the other co-authors have agreed with this change.

We hope that our manuscript is now suitable for consideration and publication in Nature Communications.

With kind regards,

Prof. Dr. Frank van Kuppeveld

Reviewers' comments:

Reviewer #1 (Remarks to the Author):

In the revised version of the manuscript, the authors modified or answered several points including the change of the title of the manuscript. The revised manuscript became clear. The authors found 1) a cell-culture adapted virus EV-D68-947 can use both Sia and sGAG as receptor, 2) EV-D68-947 can infect the cells via sGAG pathway independent of a pan-enterovirus host factor PLA2G16. Finally, they claimed that the evasion of PLA2G16 is achieved by excessive virion destabilization by sGAG. This reviewer thinks that this is the first report of a novel entry mechanism and that the two findings are important for the field of enterovirus studies.

However, the weakness of this manuscript is that the final claim by the authors is not directly proved. They would like to address the role of PLA2G16 without direct experiments on the function of PLA2G16. They provided several circumstantial evidence to support the claim. For example, 947 virus was excessively destabilized after incubation with GAG analog (Fig 4d) in vitro. As this reviewer has pointed out in the last review, in vitro uncoating can easily occur in some enteroviruses including poliovirus, but poliovirus infection is PLA2G16-dependent. This suggests that extent of destabilization in vitro cannot be a marker for PLA2G16-dependency. Secondly, in Fig2a and c, low pH-dependency and ICAM-5-dependency of 947 is different from 2042 and 2042-4/7. These pH and ICAM-5 dependencies did not correlated with PLA2G16 dependency. Third, an EV71 mutant, which is independent of PLA2G16, showed decreased thermostability (Fig 4h). However, 947 is not less thermostable compared with other Sia dependent strains (Fig 2b). Again, thermostability cannot a marker for PLA2G16-dependency. Unfortunately I was not convinced with the claim by the authors. More direct evidences are needed to support the claim.

When I see that the sGAG pathway is not clathrin-mediated, I had a feeling that PLA2G16 is not the molecule to destabilize the virion as the authors think. Do the authors have evidence to exclude the possibility that PLA2G16 prevent random uncoating, in which RNA release is not regulated and RNA can exit in any 60 gates, and that PLA2G16 navigates a correct exit so that RNA can exit across the membrane? I cannot suggest a good experiment to prove this easily. In the same context, I have another question. There is a possibility that some other molecules replace the above function of PLA2G16. Are there any lipid modifying enzymes that hit in 947 experiment but not Sia-dependent virus experiment?

Response to reviewer's comments on manuscript of Baggen et al. entitled "Bypassing pan-enterovirus host factor PLA2G16"

Previous manuscript title: "Evasion of pan-enterovirus host factor PLA2G16 via excessive virion destabilization".

Reviewer #1 (Remarks to the Author):

In the revised version of the manuscript, the authors modified or answered several points including the change of the title of the manuscript. The revised manuscript became clear. The authors found 1) a cell-culture adapted virus EV-D68-947 can use both Sia and sGAG as receptor, 2) EV-D68-947 can infect the cells via sGAG pathway independent of a pan-enterovirus host factor PLA2G16. Finally, they claimed that the evasion of PLA2G16 is achieved by excessive virion destabilization by sGAG. This reviewer thinks that this is the first report of a novel entry mechanism and that the two findings are important for the field of enterovirus studies.

However, the weakness of this manuscript is that the final claim by the authors is not directly proved. They would like to address the role of PLA2G16 without direct experiments on the function of PLA2G16.

First, we would like to thank the reviewer for the thorough evaluation of our manuscript. We appreciate that the reviewer thinks our findings are novel and important for the enterovirus field. As the reviewer correctly summarizes, we have shown that 1) EV-D68 strain 947 can engage both Sia and sGAGs as receptors and 2) that this virus does not require PLA2G16 when infecting cells via sGAGs. As a final effort, we sought an explanation for the ability of EV-D68-947 to circumvent PLA2G16 and obtained data from multiple angles that suggested excessive virion destabilization as a possible mechanism to bypass PLA2G16.

The final aim of our study was not, as is suggested by the reviewer, to "*address the role of PLA2G16*". Also, we have not stated that PLA2G16 is directly involved in virion destabilization (see remark below: "*PLA2G16 is not the molecule to destabilize the virion as the authors think.*"). Based on our findings, we merely suggest that PLA2G16 serves as an uncoating cue but do not claim to have identified its mechanisms of action. There are numerous possible mechanisms by which PLA2G16 could facilitate uncoating, as we have briefly discussed in the manuscript (**lines 264-268**).

A study addressing the precise role of PLA2G16 in infection would require dedicated experiments investigating the direct activity of PLA2G16. Instead of investigating this question, we aimed to provide a possible explanation for how EV-D68 is capable of bypassing PLA2G16. This question requires a different approach than the first question, because the mechanism by which an enterovirus bypasses PLA2G16 can be completely different from the function of PLA2G16 itself.

To make sure that it is clear to the reader that the main topic of this study is "bypassing PLA2G16" and not "PLA2G16 function", we have removed a (speculative) statement about the role of PLA2G16 from the abstract.

They provided several circumstantial evidence to support the claim. For example, 947 virus was excessively destabilized after incubation with GAG analog (Fig 4d) in vitro. As this reviewer has pointed out in the last review, in vitro uncoating can easily occur in some enteroviruses including poliovirus, but poliovirus infection is PLA2G16-dependent. This suggests that extent of destabilization in vitro cannot be a marker for PLA2G16-dependency.

In this manuscript, we show that EV-D68-947 depends on PLA2G16 when infecting cells via Sia, whereas this virus no longer requires PLA2G16 when using sGAGs. Subsequently, we propose that this could be due to the virion-destabilizing effect of sGAGs that we observed *in vitro* (and which was not observed for Sia).

The reviewer states that the extent of receptor-mediated destabilization cannot be the cause of PLA2G16 independency, since poliovirus is also destabilized by its receptor *in vitro* and yet requires PLA2G16. Indeed, the observation that EV-D68-947 uncoats upon sGAG binding *in vitro* is not unique, since many other enterovirus receptors have the same effect on their respective virus. We therefore understand why it may not be readily apparent what makes sGAGs different from other enterovirus uncoating receptors.

An important difference between sGAGs (for EV-D68-947) and poliovirus receptor (PVR) is that PVR is indispensable for the destabilization of poliovirus to enable genome release. In contrast, sGAGs are dispensable for uncoating, since EV-D68-947 can sufficiently uncoat to establish an infection in cells lacking sGAGs (via ICAM-5 and Sia). This means that sGAGs, instead of being an obligatory uncoating receptor like PVR, are an extra layer of destabilization that is not required for EV-D68-947 uncoating per se. Due to this difference, the fact that poliovirus is PLA2G16-dependent cannot be used as an argument against our hypothesis that excessive (i.e. more than necessary for full uncoating) virion destabilization causes PLA2G16 independency.

In the discussion section of the revised manuscript, we have added a sentence that should clarify to the reader that sGAGs are “extra” uncoating receptors for EV-D68-947, in contrast to the conventional enterovirus receptors (**lines 226-228**).

Secondly, in Fig2a and c, low pH-dependency and ICAM-5-dependency of 947 is different from 2042 and 2042-4/7. These pH and ICAM-5 dependencies did not correlated with PLA2G16 dependency.

In our experiments, EV-D68-947 was consistently the most PLA2G16-independent strain, as compared to the mutant 2047-4/7, even though both viruses are capable of binding sGAGs. In Figures 3a and c, we sought a possible explanation for this phenomenon and showed that EV-D68-947 differs from 2042-4/7 in its acid dependency and ICAM-5 interaction, hinting towards a link between PLA2G16 independency and receptor-mediated destabilization.

The reviewer indicates that the PLA2G16 dependencies of the three strains (947, 2042, and 2042-4/7) do not completely correlate with their acid and ICAM-5 dependencies. However, we would like to point out that such a complete correlation is not to be expected, because the main factor influencing the PLA2G16 dependency of a strain is clearly its capacity to bind sGAGs (Fig. 2e). Since EV-D68-2042 cannot engage sGAGs and infects cells exclusively via Sia, one should not consider this strain when searching for a correlation between the dependencies on acid, ICAM-5, and PLA2G16.

Third, an EV71 mutant, which is independent of PLA2G16, showed decreased thermostability (Fig 4h). However, 947 is not less thermostable compared with other Sia dependent strains (Fig 2b). Again, thermostability cannot a marker for PLA2G16-dependency.

Our experiments with EV-D68-947 revealed a correlation between the PLA2G16 independency of this virus (via the sGAG route) and a strong destabilizing effect of a receptor (sGAGs) on the virion. This led to the hypothesis that excessive destabilization of the virion could be the reason why PLA2G16 is no longer required for infection. In the first revised version of our manuscript, we included a different enterovirus species (serotype EV-A71) to broaden the scope of this study and to strengthen our final

hypothesis with complementary data. The experiments with EV-A71 revealed a correlation between PLA2G16 independency and a reduced thermostability of the virus (**Fig. 4g,h**). These results provided an additional line of support for our hypothesis PLA2G16 independency might be enabled by excessive virion destabilization.

The reviewer points out that thermostability cannot be a marker for PLA2G16 dependency, because the thermostability of EV-D68-947 does not differ from that of the other EV-D68 strains (**Fig. 3b**). In response to this remark, we would like to emphasize that we never suggested that all enteroviruses should have a reduced thermostability in order to bypass PLA2G16. Instead, we proposed that the mechanism underlying PLA2G16 independency is “excessive virion destabilization”, which is a general description of a phenomenon that can be caused by different mechanisms.

In the case of EV-D68-947, this excessive destabilization is achieved by binding to virion-destabilizing sGAGs, whereas the PLA2G16-independent EV-A71 mutant is inherently destabilized by capsid mutations that reduce its thermostability. In the end, the engagement of an extra uncoating receptor (sGAGs) or the acquisition of destabilizing capsid mutations would lead to the same final result: an excessively destabilized particle. Therefore, our observations obtained from different enterovirus species all support the proposed link between excessive virion destabilization and PLA2G16 independency.

Unfortunately I was not convinced with the claim by the authors. More direct evidences are needed to support the claim.

The reviewer indicates that we should provide more direct evidence to support our claims. Our hypothesis that enteroviruses might bypass PLA2G16 via excessive virion destabilization is based on correlations between PLA2G16 independency and virion instability (achieved by different mechanisms) that we have observed for EV-D68-947 and EV-A71. We agree with the reviewer that it would be satisfying to have direct evidence for a causal relationship between virion stability and PLA2G16 dependency.

To obtain such direct evidence, one would have to measure PLA2G16 dependency outside the context of a cell, in order to rule out effects of other (unknown) factors on PLA2G16 dependency. Unfortunately, such a cell-free assay to measure PLA2G16 dependency does not yet exist and would require the *in vitro* generation of membrane systems that contain uncoating receptors as well as PLA2G16, followed by monitoring pore formation and RNA translocation from receptor-bound virions. The method that best approaches such an assay (Lee, H. *et al.* Science Advances 1–10, 2016) employs lipid bilayer nanodiscs with embedded receptors, combined with cryo-EM, and has provided valuable first insights into the asymmetric uncoating of a membrane-bound enterovirus. This approach has been a technical challenge and has not yet reached a sufficient local resolution to visualize the transmembrane pore and the translocating genomic RNA. Although it would theoretically be possible to improve this method and to include an active PLA2G16 protein, the development of an *in vitro* assay to study PLA2G16 functionality would be a breakthrough in the field in itself. Therefore, we currently rely on measuring PLA2G16 dependencies in infected cells and trying to correlate PLA2G16 independency with other properties of a virus.

To ensure that the reader does not expect that we present direct evidence for a causal link between virion stability and PLA2G16 independency, we have reformulated the title of the revised manuscript in a more cautious manner, by omitting the proposed mechanisms by which PLA2G16 is bypassed;

“Evasion of pan-enterovirus host factor PLA2G16 via excessive virion destabilization” was changed to “Bypassing pan-enterovirus host factor PLA2G16”.

When I see that the sGAG pathway is not clathrin-mediated, I had a feeling that PLA2G16 is not the molecule to destabilize the virion as the authors think.

The reviewer believes it is unlikely that PLA2G16 destabilizes the virus by directly interacting with the virion. As indicated earlier, we have not suggested that PLA2G16 directly interacts with the virion, but we merely propose that PLA2G16 serves as an uncoating cue, a function that it may fulfil via various possible mechanisms.

We agree with the reviewer that PLA2G16 is most likely to perform this function via an indirect mechanism and this possibility was already mentioned in the discussion section of the manuscript (**line 267**).

Do the authors have evidence to exclude the possibility that PLA2G16 prevent random uncoating, in which RNA release is not regulated and RNA can exit in any 60 gates, and that PLA2G16 navigates a correct exit so that RNA can exit across the membrane? I cannot suggest a good experiment to prove this easily.

The reviewer hypothesizes that the function of PLA2G16 might be to steer the release of RNA in the direction of a capsid opening that faces the endosomal membrane, thereby preventing RNA release into the endosomal lumen. This scenario fits very well in our hypothesis (that PLA2G16 serves as an uncoating cue), since regulating the directionality of RNA release is one of many ways in which PLA2G16 could facilitate correct uncoating. We therefore have no reason to exclude this possibility.

To provide the reader with a brief summary of the possible mechanistic roles of PLA2G16, we have added a sentence to the discussion section of the revised manuscript (**lines 264-266**).

In the same context, I have another question. There is a possibility that some other molecules replace the above function of PLA2G16. Are there any lipid modifying enzymes that hit in 947 experiment but not Sia-dependent virus experiment?

The EV-D68-947 screen identified two genes encoding lipid-modifying factors that were not enriched in the EV-D68-Fermon screen: *SACMIL* and *FAAH2*. However, we did not investigate these factors because these genes were also significantly enriched in a published haploid screen with an antibody that detects heparan sulfate (Jae, L. T. *et al.* Science 340, 479–484, 2013), indicating that these genes are required for expression of sGAGs on the cell surface. No other lipid-modifying enzymes were among the most significantly enriched hits in the EV-D68-947 screen.

We hope that our manuscript is now considered suitable for acceptance and publication in Nature Communications.

With kind regards,

Prof. Dr. Frank van Kuppeveld

REVIEWERS' COMMENTS:

Reviewer #1 (Remarks to the Author):

I read the rebuttal by the authors carefully. I understand why it was difficult for me to agree with the authors.

The authors found that a cell culture adapted EV-D68-947 strain can use both Sia and sGAG as receptors. The virus can infect the cells via sGAG pathway independent of a pan-enterovirus host factor PLA2G16. The experimental data are very clear and convincing. The authors have a hypothesis this evasion of PLA2G16 is due to the "excessive virion destabilization" by sGAG. I had an impression that the authors think that the PLA2G16 contributes to destabilization of the virion and that excessive destabilization of the virion by GAG can compensate the destabilization effect by PLA2G16. I think that the authors should clarify what is "excessive destabilization" or "extra layer of destabilization".

When I reconsider the explanation of the results, I came up with a new explanation. The virion of EV-D68-947 can be destabilized via pathway different from canonical uncoating pathway mediated by Sia and PLA2G16. The mutations introduced in this virus allow different 3D structures which can bind sGAG and undergo different pathway. Supplemental Fig. 1c suggests that the infection via sGAG pathway does not require endocytosis. In addition, in Fig.3a and supplementary Fig 3a, infection via sGAG pathway does not require acidification of the endosome. In Fig. 3c, infection of 947 strain was inhibited by soluble ICAM-5, suggesting that the 947 strain was uncoated in vitro by the soluble ICAM-5 at neutral pH before the virus meets the cells. Empty particles were produced by incubation with dp6 and LMWH at neutral pH. These results together supported the notion that 947 strain can be destabilized in the absence of low pH. The uncoating of 947 strain by sGAG pathway can occur at the cell surface. This pathway may not require PLA2G16 function probably because the uncoating completes before the entry of the virion in the cells and before the virion is incorporated into autophagosome. If this is correct, the main reason for bypassing PLA2G16 is not excessive destabilization but low pH-independent uncoating. I suggest to rewrite the manuscript based on this idea.

Response to reviewer's comments on manuscript of Baggen et al. entitled "Bypassing pan-enterovirus host factor PLA2G16"

REVIEWERS' COMMENTS:

Reviewer #1 (Remarks to the Author):

I read the rebuttal by the authors carefully. I understand why it was difficult for me to agree with the authors. The authors found that a cell culture adapted EV-D68-947 strain can use both Sia and sGAG as receptors. The virus can infect the cells via sGAG pathway independent of a pan-enterovirus host factor PLA2G16. The experimental data are very clear and convincing. The authors have a hypothesis this evasion of PLA2G16 is due to the "excessive virion destabilization" by sGAG. I had an impression that the authors think that the PLA2G16 contributes to destabilization of the virion and that excessive destabilization of the virion by GAG can compensate the destabilization effect by PLA2G16. I think that the authors should clarify what is "excessive destabilization" or "extra layer of destabilization".

The reviewer did not raise any concerns regarding the issues discussed in the previous two rebuttals, so it appears that we addressed all questions raised by this reviewer properly.

Regarding the request to further clarify the term "excessive": EV-D68 uses Sia as receptor (together with ICAM-5) and is, like all other enteroviruses, dependent on host factor PLA2G16. We previously showed that EV-D68-947 can infect cells in the absence of Sia receptors. In this study, we demonstrate that this virus then uses sGAGs as receptor and at the same time becomes PLA2G16 independent. We showed that sGAGs are a destabilizing cue, as EM studies revealed that sGAGs (and not Sia) induce in vitro genome release from EV-D68-947 particles (i.e. destabilization). sGAGs are not a strict requirement for uncoating, since EV-D68-947 can still infect cells lacking sGAGs (via ICAM-5, Sia and PLA2G16). Thus, because sGAGs are a destabilizing force that is dispensable for uncoating, we described sGAGs as an "extra layer of destabilization" that induces "excessive destabilization". When sGAGs are used as receptor, this "excessive destabilization" may allow the virus to bypass PLA2G16. In more general terms, according to our model the collective destabilizing effect of different uncoating cues is needed to prime the virion for PLA2G16-mediated infection. If the virion is destabilized more than is required for this priming (excessive destabilization), PLA2G16 could become redundant.

To further clarify this issue in the manuscript, we have added a more elaborate explanation (lines 205-207) indicating that "excessive destabilization" refers to "more extensive destabilization than is minimally required to prime the virion for PLA2G16-mediated infection".

When I reconsider the explanation of the results, I came up with a new explanation. The virion of EV-D68-947 can be destabilized via pathway different from canonical uncoating pathway mediated by Sia and PLA2G16. The mutations introduced in this virus allow different 3D structures which can bind sGAG and undergo different pathway. Supplemental Fig. 1c suggests that the infection via sGAG pathway does not require endocytosis. In addition, in Fig.3a and supplementary Fig 3a, infection via sGAG pathway does not require acidification of the endosome. In Fig. 3c, infection of 947 strain was inhibited by soluble ICAM-5, suggesting that the 947 strain was uncoated in vitro by the soluble ICAM-5 at neutral pH before the virus meets the cells. Empty particles were produced by incubation with dp6 and LMWH at neutral pH. These results together supported the notion that 947 strain can be destabilized in the absence of low pH. The uncoating of 947 strain by sGAG pathway can occur at the cell surface. This pathway may not require PLA2G16 function probably because the uncoating completes before the entry of the virion in the cells and before the virion is incorporated into autophagosome. If this is correct, the main reason for bypassing PLA2G16 is not excessive destabilization but low pH-independent uncoating. I suggest to rewrite the manuscript based on this idea.

The reviewer now comes up with a new explanation for our results, by hypothesizing that the bypassing of PLA2G16 is not due to excessive virion destabilization, but due to low pH-independent uncoating instead. However, our data show that low pH-independent uncoating is not unique to the sGAG pathway, since EV-D68-947 uncoats independently of low pH when using the sGAG route as well as the Sia route (Supplementary figure 3b). EV-D68-947 is able to bypass PLA2G16 only when using the sGAG route (Figure 2b). Therefore, the fact that EV-D68-947 uncoats independently of low pH cannot be the explanation for its capacity to bypass PLA2G16.

To indicate more clearly in the manuscript that the pH independency of EV-D68-947 is not a consequence of using a specific glycan receptor, we have addressed this point more elaborately in the results section (lines 152-154).

We hope that our manuscript is now considered suitable for acceptance and publication in Nature Communications.

With kind regards,

Prof. Dr. Frank van Kuppeveld